# Insights into the tectonic evolution of the Svecofennian orogeny based on *in situ* Lu-Hf dating of garnet and apatite from Olkiluoto, SW Finland

**JON ENGSTRÖM [1,2*], KATHRYN CUTTS[1], STIJN GLORIE[3] ESA HEILIMO[4] ESTER M. JOLIS[1] AND RADOSLAW M. MICHALLIK[1]**

[1]*Geological Survey of Finland, P.O. Box 96, FI-02151 Espoo Finland.*

[2]*Åbo Akademi University, Akatemiakatu 1, 20500 Turku*

[3]*Department of Earth Sciences, University of Adelaide, Adelaide, SA 5005, Australia*

[4]*University of Turku, Akatemiakatu 1, 20500 Turku*

\* Corresponding author (e-mail: jon.engstrom@gtk.fi)

**Abstract.** The southern Finland granites and associated migmatitic rocks have a complex metamorphic history, being affected by multiple events during the ca. 1.88-1.79 Ga Svecofennian orogeny. In this study, the prolonged tectonic evolution of migmatites and associated rocks in southwestern Finland has been investigated using the new *in situ* Lu-Hf method. Results reveal detailed temporal constraints for the tectonic evolution that can be linked to major events in adjacent tectonic blocks in both Finland and Sweden during the Svecofennian orogeny. The metamorphic peak at the Olkiluoto site occurred at 1829 ± 19 Ma based on *in situ* Lu-Hf dating of garnet. The pressure-temperature (P-T) path for the rocks indicates a prograde evolution, with peak P-T conditions of 3-5 kbar and approximately 700 °C. Younger ages of ca. 1780 Ma obtained using both Lu-Hf and U-Pb systems in apatite inclusions in garnet indicate rapid cooling at 1780 Ma. Based on the metamorphic constraints and obtained ages, we link the Olkiluoto site to the Häme orogenic belt in southern Finland and also support the proposed thermal-tectonic connection with the Ljusdal lithotectonic unit in central Sweden.

## 1 Introduction

The tectonic evolution and the metamorphic record from the latter parts of Svecofennian orogeny has in the past been inferred to be similar in southern Finland and central Sweden (Hietanen, 1975; Korja and Heikkinen, 2005; Högdahl and Bergman, 2020; Engström et al., 2022). However, the coupling between these regions has been challenging to establish due to the prolonged tectonic evolution and the polymetamorphic nature of the Palaeoproterozoic bedrock, which often obscures the detailed P-T-t record for these rocks. Thus, novel geochronological techniques, particularly *in situ* Lu-Hf geochronology focusing on key metamorphic minerals like garnet and apatite, are indispensable for accurately delineating and investigating these ancient polymetamorphic terranes, especially since they retain information of past metamorphism in rocks that have seen multiple tectonic events, that is typical for rocks in ancient orogenies (Tamblyn et al., 2022). Dating garnet, coupled with conventional geochronological techniques such as U-Pb geochronology on accessory zircons and monazite is beneficial because garnet as a metamorphic mineral clearly represents the metamorphic age of the rock and garnet can also provide information about the P-T evolution of the rock. (Brown et al., 2022; Tamblyn et al., 2022; Simpson et al., 2023). Using garnet and apatite Lu-Hf geochronology in tandem is a powerful tool for understanding the thermal evolution of metamorphic rocks due to the difference in closure temperature of the Lu-Hf system in garnet (>800 °C; Smit et al., 2024) and apatite (670-730 °C; Glorie et al., 2024a), this is essential information for understanding polymetamorphic rocks.

This study focuses on resolving the tectonic evolution in the Olkiluoto study area, which was affected by two metamorphic events during the Palaeoproterozoic Svecofennian orogeny (Tuisku and Kärki, 2010; Saukko et al., 2020). The Palaeoproterozoic bedrock of southern Finland consists to a large extent of granitoids and migmatites (e.g. Nironen, 2017). Based on lithological, geochemical and geochronological data, the Svecofennian crustal domain in Finland is divided into two major lithotectonic units: the Western Finland Subprovince (WFS); and the Southern

Finland Subprovince (SFS) (Fig. 1; Korsman et al., 1997; Väisänen et al., 2002; Lahtinen et al., 2005; Nironen, 2017). The Svecofennian Province of Finland is separated from the Ljusdal lithotectonic unit in central Sweden (Högdahl and Bergman, 2020) by the Gulf of Bothnia. Despite this geographical separation, both regions show similar characteristics in terms of the rock types and structures (Fig. 1). These similar characteristics are comparable magmatic activity and similar structural evolution coupled with the same style and timing of metamorphism (e.g. Kähkönen, 2005; Nironen, 2005; Bergman et al., 2008; Väisänen et al., 2012; Högdahl and Bergman, 2020).

The Olkiluoto site is the location for the Finnish deep geological repository for spent nuclear fuel, and this study is part of the geological site characterisation. Our investigation provides new insights into the tectonic history of the Olkiluoto site and southwestern Finland during the Palaeoproterozoic Svecofennian orogeny (Fig. 1). The tectonic evolution has been defined through the analysis of garnet, a key mineral that serves as a reliable indicator of metamorphic conditions and thermal history within the crust. We have used the recently developed *in situ* Lu-Hf geochronology employing the use of laser ablation-inductively coupled plasma tandem-quadrupole-mass spectrometry (LA-ICP-Q-MS/MS) (Brown et al., 2022; Simpson et al., 2021, 2023) to demonstrate that garnet and also apatite in felsic, migmatitic tonalitic-granitic-granodioritic (TGG) intrusive rocks show metamorphism with one distinct metamorphic event and possibly an earlier event. Even though the Olkiluoto investigation area is small, the results of this study can be connected to a more regional context regarding the tectonic framework in southern Finland. The Lu-Hf geochronology from the Olkiluoto site combined with pressure-temperature modelling, provide new insight into how metamorphic processes and tectonic events were interconnected in southern Finland. This knowledge is important for establishing connections with the Ljusdal lithotectonic unit in central Sweden. Recent studies by Engström et al. (2022); Lahtinen et al. (2023); Luth et al. (2024) infer that the coupling of the Olkiluoto area to central Sweden is plausible. However, more constraints and detailed research is required from adjacent areas in southwestern Finland and central Sweden to define the tectonic and metamorphic evolution and the coupling between these two areas.

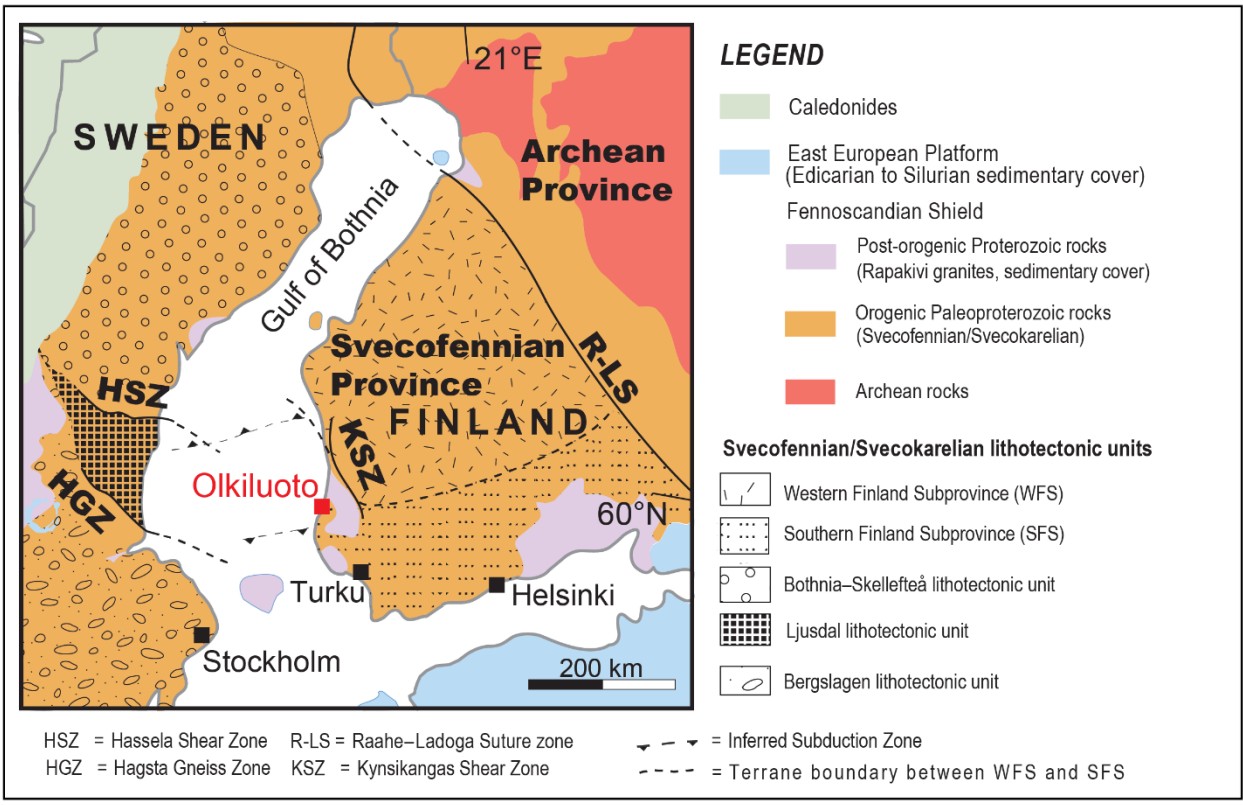

**Figure 1.** Synthetic geological map of the Fennoscandian shield. Olkiluoto is indicated with a red square. Map modified from Koistinen et al., 2001; Korja and Heikkinen, 2005; Nironen, 2017 and Stephens, 2020.

## 2 Geological setting of the study area

### 2.1 Tectonic framework

The Palaeoproterozoic Svecofennian orogeny, and its corresponding crustal province, were first introduced in the classic review by Gaál and Gorbatschev (1987) (Fig. 1). Since then, several tectonic models have been presented for the evolution of the accretionary orogen (e.g. Nironen, 1997, Lahtinen et al., 2005, Lahtinen et al., 2023). The orogeny initiated from 1.92 Ga to 1.87 Ga (Nironen, 2017; Heilimo et al., 2023) with a collisional stage during which several volcanic arc complexes or microcontinents laterally accreted onto the margin of the Archean Karelia craton (e.g. Lahtinen et al., 2005). The Svecofennian orogeny is characterized by two main high-T/low-P type metamorphic events. The first event occurred at 1.88–1.87 Ga, reaching upper amphibolite facies, and can be detected throughout the Finnish Svecofennian (Korsman et al., 1999; Nironen, 2017). The convergence stage included several thrust sheets that developed within a W–SW to E–NE compressional environment in southern Finland (Nironen, 2017; Torvela & Kurhila, 2020). The Western Finland Subprovince (WFS) was subjected to moderate crustal thickening, and the widespread development of granites and associated migmatites related to either channelized flow or in-situ melting of metasedimentary rocks, with peak metamorphism occurring at 1.88-1.87 Ga (Mäkitie et al., 2012; Chopin et al., 2020). The subsequent tectonic phases included minor crustal extension, followed by the resumption of orogenic convergence that resumed at ca. 1.84 Ga and especially in southern Finland initiated a younger metamorphic event forming granites and associated migmatites (Lahtinen et al., 2005; Torvela et al., 2008; Torvela and Kurhila, 2020; Kara et al., 2021). The younger metamorphic event involved high-T metamorphism to granulite facies in large areas of southernmost Finland and was associated with emplacement of granites together with anatectic melting resulting in the formation of migmatites and pegmatites during the late stages of the Svecofennian orogeny (Korsman et al., 1999; Väisänen and Hölttä, 1999; Väisänen et al., 2002; Skyttä and Mänttäri, 2008).  This transpressional deformation phase was

characterized by intensive folding and shear zone development (Ehlers et al., 1993; Torvela and Kurhila, 2020; Väisänen et al., 2002; Väisänen and Skyttä, 2007).

The Svecofennian rocks within the Ljusdal lithotectonic unit, central Sweden were intruded by the Ljusdal Batholith at ca. 1.86-1.84 Ga (Högdahl et al., 2008) and were affected by polyphase, ductile deformation, coupled to two episodes of high-grade, low-pressure metamorphism during the Svecofennian orogeny, dated at ca. 1.85 Ga and 1.83–1.82 Ga, with prolonged crustal heating continuing to at least 1.80 Ga (Högdahl et al., 2008; Högdahl and Bergman, 2020, see Fig. 1). Especially, the latter metamorphic events in central Sweden (1.83–1.82 Ga) and southern Finland (1.84–1.80 Ga) show similar tectonic evolution and metamorphic signatures, thus a connection between the Olkiluoto site and the Ljusdal lithotectonic unit in central Sweden can be deduced (Högdahl et al., 2008; Engström et al., 2022).

## 2.2   Geology of the Olkiluoto area

According to Engström et al. (2022), the Olkiluoto site experienced a tectonic evolution where ductile deformation took place in several steps, coinciding with the formation of migmatites and leucosomes under high-T conditions in the Palaeoproterozoic crust. Thus, the bedrock at Olkiluoto island consists of Palaeoproterozoic, mostly intrusive and supracrustal (meta-pelites, meta-arenites and meta-volcanic) rocks and is situated in the westernmost part of the Southern Finland Subprovince (SFS) (Fig. 1 and Fig. 2). The felsic, tonalitic-granitic-granodioritic (TGG) intrusive rocks are migmatised with small, injected veins of pegmatitic-granitic (PGR) leucosome. Since this TGG intrusive rock is less deformed and altered by the subsequent polyphase ductile deformation events compared to the metapelitic supracrustal migmatitic rocks (Engström et al., 2022), it is well suited for our study on the metamorphic evolution in Olkiluoto. The whole bedrock at the site is also intruded by diabase dykes, likely of Mesoproterozoic age (Mänttäri et al., 2006). The migmatites in Olkiluoto are divided into two main groups: vein- and dyke-structured metatexites (VGN in Fig. 2); and nebulitic diatexites (DGN in Fig. 2), which can be further subdivided into several subtypes on the basis of their migmatite structures (Kärki, 2015). Metatexitic migmatites dominate the northwest part of the island, whereas diatexites are abundant in the southeast part of the island (Fig. 2).

Earlier studies indicate that two distinct metamorphic events occurred in Olkiluoto (Tuisku and Kärki, 2010; Saukko et al., 2020; Engström et al., 2022), with the metamorphic conditions of the first event (M1, older) estimated to have a peak pressure of approximately 6 kbar, but no information on temperature. This M1 event was interpreted based on some samples giving a higher estimated peak pressure than the average metamorphic grade (3-4 kbar), this event was interpreted to be early and connected to magmatic processes and emplacement of the protolith of TGG rocks (Tuisku and Kärki, 2010). Even though the earlier metamorphic studies are lacking the temperature information, the study by Saukko et al. (2020), concluded that two generations of migmatitic events with leucosome production did occur in Olkiluoto. The mineral assemblages of the second metamorphic peak (M2, younger) are indicative of upper amphibolite facies, stable at 660–700 ºC and 3.7 – 4.2 kbar (Tuisku and Kärki, 2010). The timing of these tectono-metamorphic events are constrained using tectonic events and metamorphic U-Pb zircon ages at ca. 1.87–1.84 Ga and 1.82–1.78 Ga, respectively (Engström et al., 2022). The pressure difference of approximately 2 kbar between the two metamorphic stages indicate either an erosion phase between the metamorphic phases or a significant crustal uplift, another possibility is an interplay between both processes. The M2, metamorphic event is characterized by injected granitic and pegmatitic leucosome veins and dykes that are crosscutting the earlier generated foliation (Engström et al., 2022).

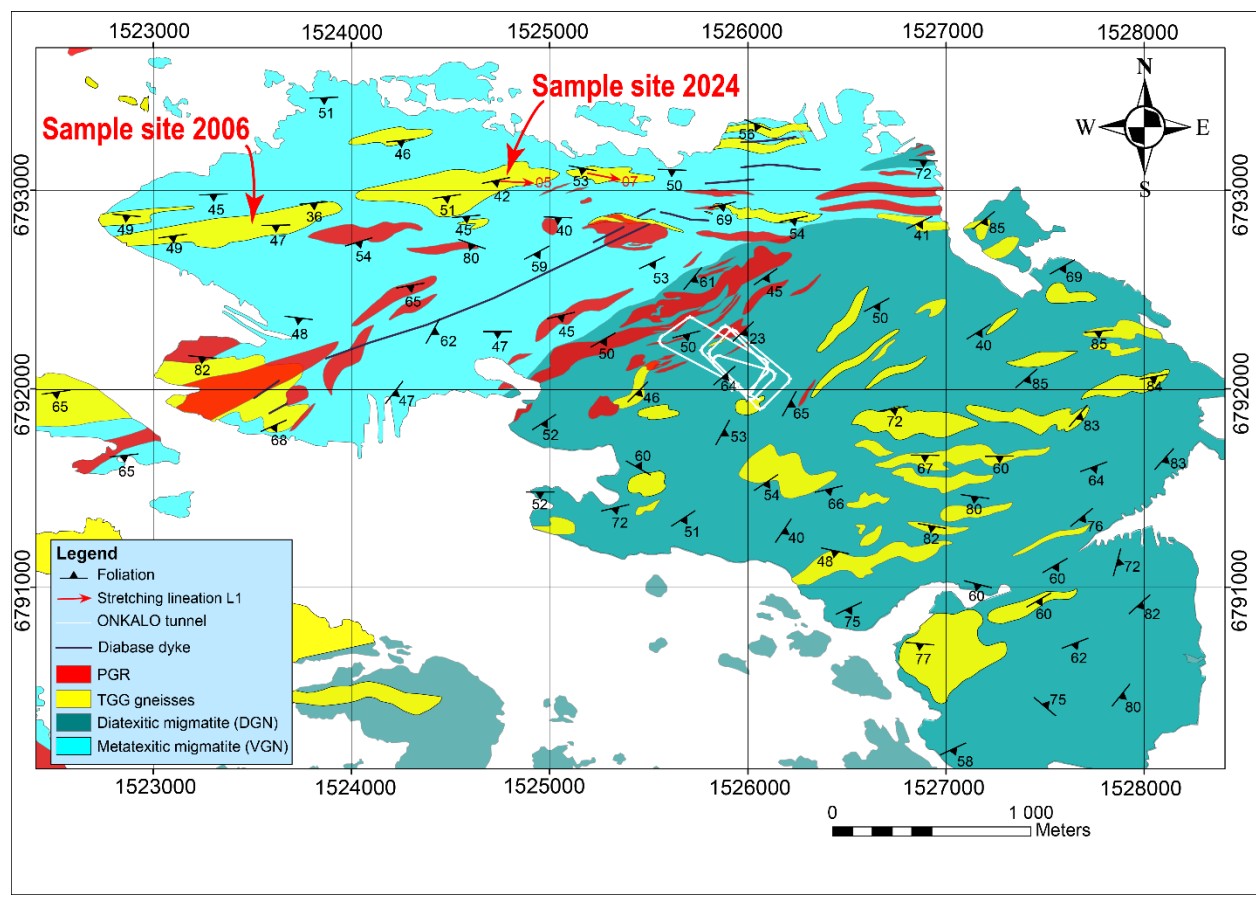

**Figure 2.** Geological map of Olkiluoto (modified from Aaltonen et al., 2016 and Engström et al., 2022). The locations of the investigated outcrops in 2024 and 2006 are also indicated. Coordinate system in the map Finnish Projected KKJ Zone 1.

135

## 3    Methodology

### 3.1 Outcrop description

This study includes whole rock geochemistry of the different lithological units at the site coupled with a detailed outcrop study of the TGG intrusive rock representing the protolith for the metamorphic rock in the M1 metamorphic

140    phase in Olkiluoto (Engström et al., 2022). Detailed structural geological mapping of the TGG outcrop was performed to guide the sampling for garnets (see Fig. 2, sample site 2024). From the studied outcrop (see Fig. 2, sample site 2024) two rock samples (approx. 50 cm x 10 cm x 8 cm) were collected for further investigation (Fig. 3). The samples MM30 (predominantly small garnets, Fig, 3A-B) and MM31 (predominantly big garnets, Fig. 3C-D) were chosen carefully during the mapping to have a structural control since the big garnets in MM31 were observed with the

145    leucosome on the outcrop. The samples are separated by 3-4 m and during detailed investigation are shown to be compositionally similar, with both containing large subhedral garnet crystals up to 3-5 cm in size in leucosomes and small (up to 0.3-0.7 cm in size) anhedral garnet crystals within the mesosome. Garnet grains were selected from these samples for microanalytical and petrochronological analysis, such as thin sections and micro-XRF images. Both large (MM31) and small (MM30) garnets were targeted for Lu-Hf analysis. Apatite inclusions in the large garnet (MM31)

150    were targeted for Lu-Hf and U-Pb analysis.

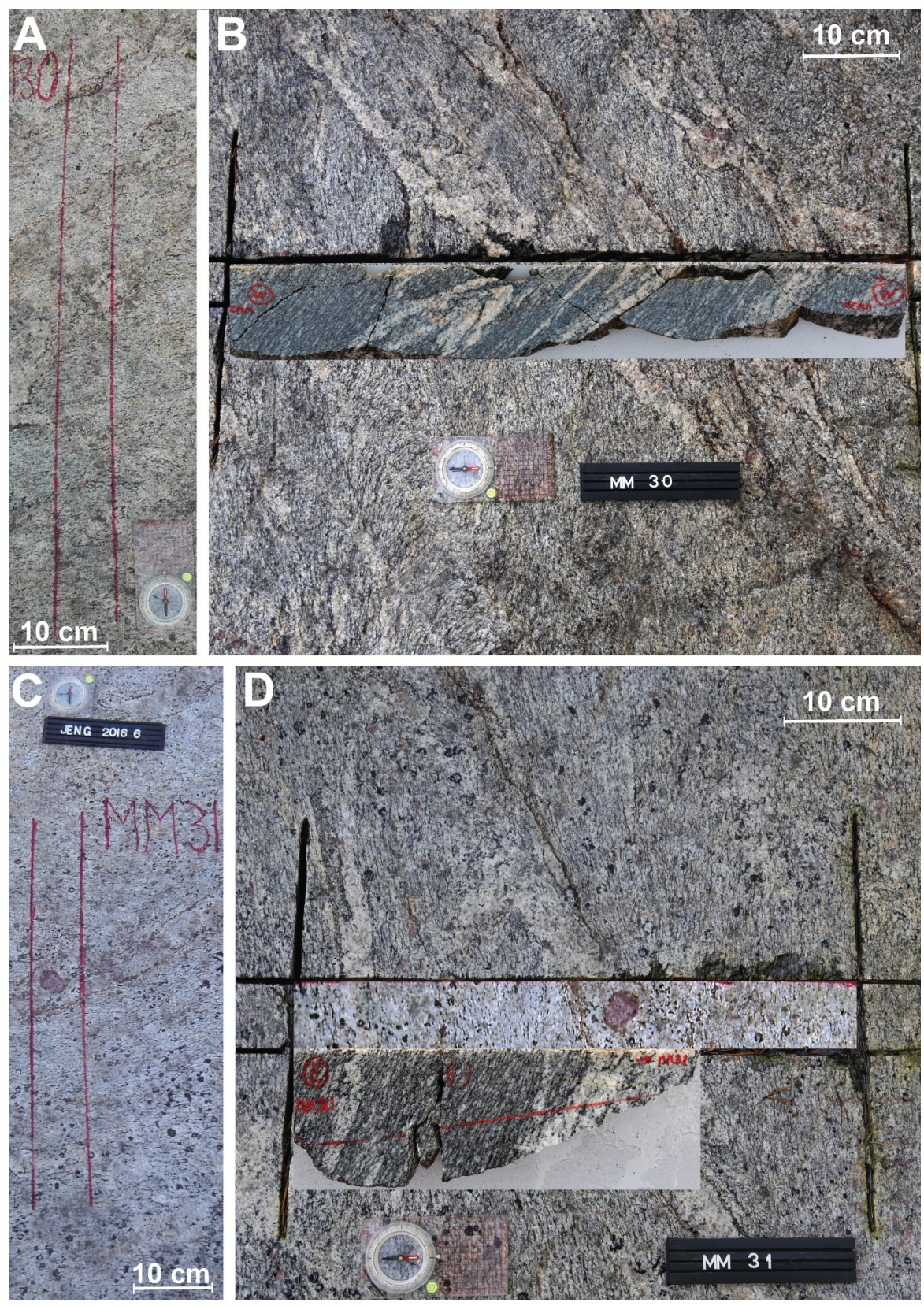

**Figure 3.** The TGG outcrop indicating the sawed-out rock sample areas. Sample area MM30, with view from the top (A) and with cut-out (B). Sample area MM31, with view from the top (C) and with cut-out (D).

## 3.2 Sample description

The TGG bedrock in the northwestern part of the Olkiluoto site was chosen for this study, since it has been least affected by the different polyphase ductile deformation events, especially concerning later folding and migmatitation events. The rocks are pale-grey in colour and contain a compositional banding with a prominent stretching lineation of feldspars. Garnet occurs as small grains scattered in the mesosome (sample MM30) and as large grains that occur within leucosomes (sample MM31). Both types of garnet were targeted in this study to determine if they grew during single or multiple phases of metamorphism. The small garnet grains (up to 0.7 cm, MM30) (Fig. 3A-B) were removed from one of the collected samples by slicing it with a small saw and then picking out the small grains where they were observed. Three small grains were then mounted in a single 2.5 cm epoxy mount. The large garnet grain was embedded in the leucosome (ca. 5 cm) of sample MM31 (Fig. 3C-D) and was removed by sawing and then cut in half in order to fit the 2.5 cm epoxy mount. The mineralogy of the rock samples consists of quartz, K-feldspar, plagioclase, biotite and garnet, with accessory apatite and cordierite, which often exhibit pinite alteration (Fig. 4). The samples are coarse grained, with K-feldspar grains up to 1 cm and plagioclase and quartz of up to 0.5 cm (Fig. 4B-C). The grain size in the leucosome is larger, with K-feldspar, quartz and garnet up to several cm in size. Garnet grains in both the mesosome and leucosomes have cores rich in quartz inclusions, often define a symplectitic-like texture (Fig. 4D). Rarely, biotite is also observed in garnet cores. Apatite inclusions commonly occur on the outer edge of the core domain and in the rims (Fig. 5). Garnet rims are generally inclusion-poor, but where inclusions occur, they are large and usually consist of quartz, apatite, biotite or K-feldspar. Garnet grains are subhedral with irregular grain boundaries, often with embayments (Fig. 4D). The mesosome foliation is defined by biotite, which forms elongate grains up to several mm in size. Biotite is generally also slightly coarser in the leucosomes and the foliation is not as well defined with biotite grains often wrapping around large garnet grains. In both the mesosome and the leucosome, biotite grains are subhedral, often having scalloped grain edges. K-feldspar, quartz and plagioclase all have irregular grain boundaries, with scalloped edges, they occur as rounded inclusions in each other, and form thin, film-like segregations (Fig. 4B-D). Larger grains are often elongate and oriented parallel to the foliation of the sample. Apatite is a common accessory mineral in the mesosome of the sample, often it occurs as clumps of grains adjacent to large garnet grains but also as isolated grains with biotite or on feldspar and quartz grain boundaries.

## 3.3   Whole-rock geochemistry

All the sample preparation and analytical work for the whole-rock chemical analyses samples were carried out in the SGS Minerals Services laboratory, Canada (Kärki and Paulamäki, 2006). Rock hand specimens (0.5–2 kg) were crushed by magnesium steel jaw crusher and then pulverized in carbon steel bowl. The analyses were done by X-ray fluorescence (XRF) analyser, neutron activation analyser (NAA), inductively coupled plasma atomic emission analyser (ICP), inductively coupled plasma mass spectrometer (ICP-MS), sulphur and carbon analyser (LECO) and by using ion specific electrodes (ISE).

## 3.4   Mineral chemistry

Another large garnet from the sample MM31 was prepared as a thin section and several regions with smaller garnets were also prepared as thin sections. The garnet epoxy mounts were imaged (Fig. 5) using a Bruker Micro-XRF M4 Tornado hosted at the Geological Survey of Finland (GTK). The system is equipped with a 30-Watt rhodium (Rh) anode X-ray tube, two 30 mm$^2$ silicon drift detectors (SDD) with an energy resolution of < 145 eV (MnKα) at 275 kcps (kilocounts per second) via beryllium windows and poly-capillary optics. All data acquisition was performed with an accelerating voltage of 50 kV, a beam current of 500 µA using a fixed spot size of 20 µm under a 2 mbar vacuum. The samples were measured in one single run using a step size of 40 µm and a pixel dwell time of 20 ms/pixel. The qualitative elemental maps were generated using the Bruker M4 software with later processing in XMapTools (Lanari et al., 2014). Quantified chemical analysis (Supplementary Tab. 1) of major minerals (garnet, biotite, K-feldspar and plagioclase) using the thin section samples were obtained using a CAMECA SX100 electron microprobe analyzer (EMPA) at GTK using the WDS (wavelength-dispersive) technique. Accelerating voltage and beam current

were set to 15kV and 15nA, respectively. A defocused beam diameter of 5 µm was used for the spot analysis. Analytical results have been corrected using the PAP on-line correction program (Pouchou and Pichoir, 1986). Natural minerals and synthetic metals were used as standards. All mineral chemical data is included in the Supplementary Tab. 2 and 3.

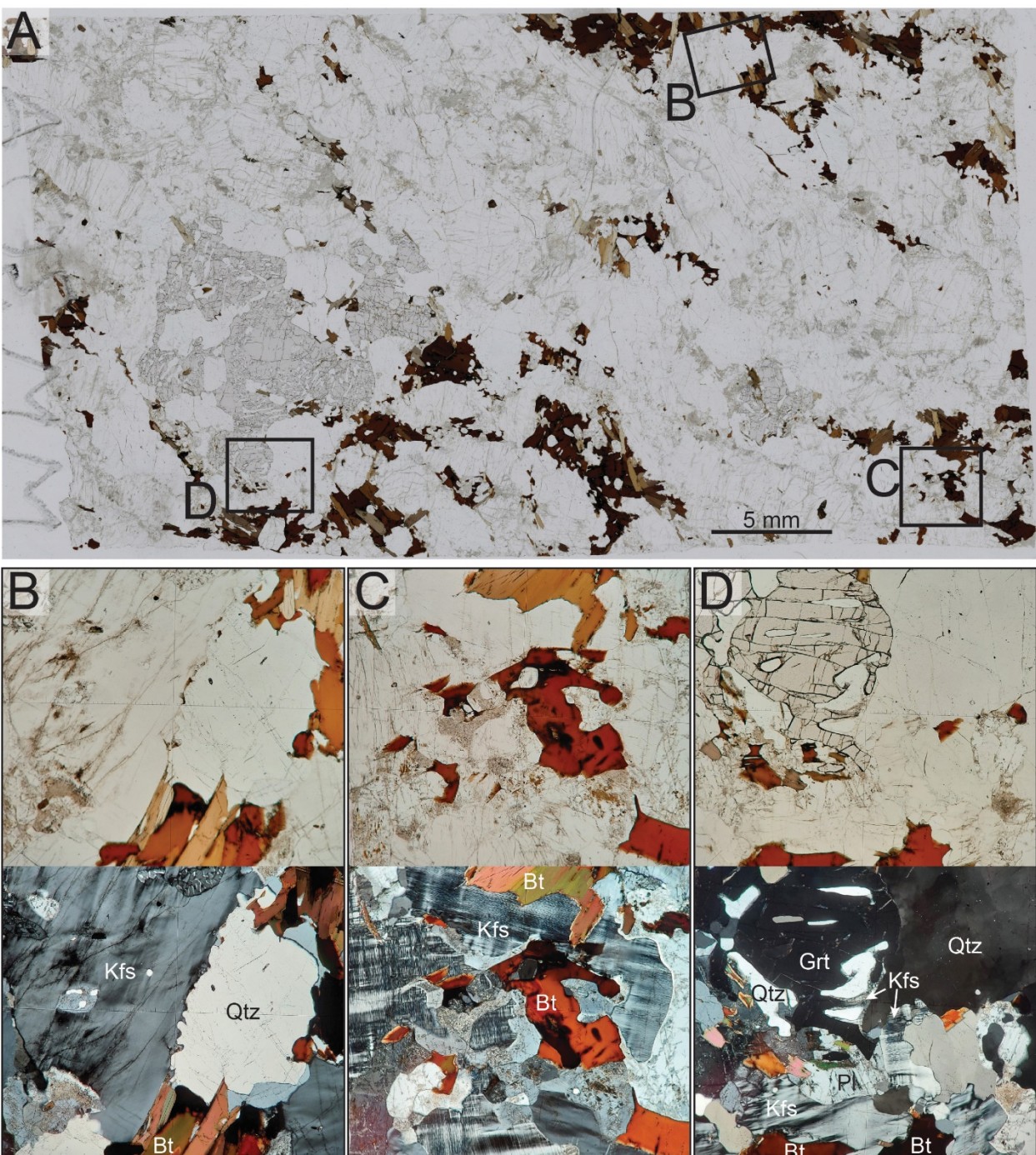

**Figure 4.** Sample photomicrographs. A. Plane polarised image of thin section for sample MM30 with the location of images shown in B-D as plane and cross polarised images. B. K-feldspar and quartz grains showing irregular grain boundaries. C. Biotite grains with scalloped edges, finer grained intergrowths of K-feldspar and plagioclase. D. Edge of garnet grain with thin films of K-feldspar.

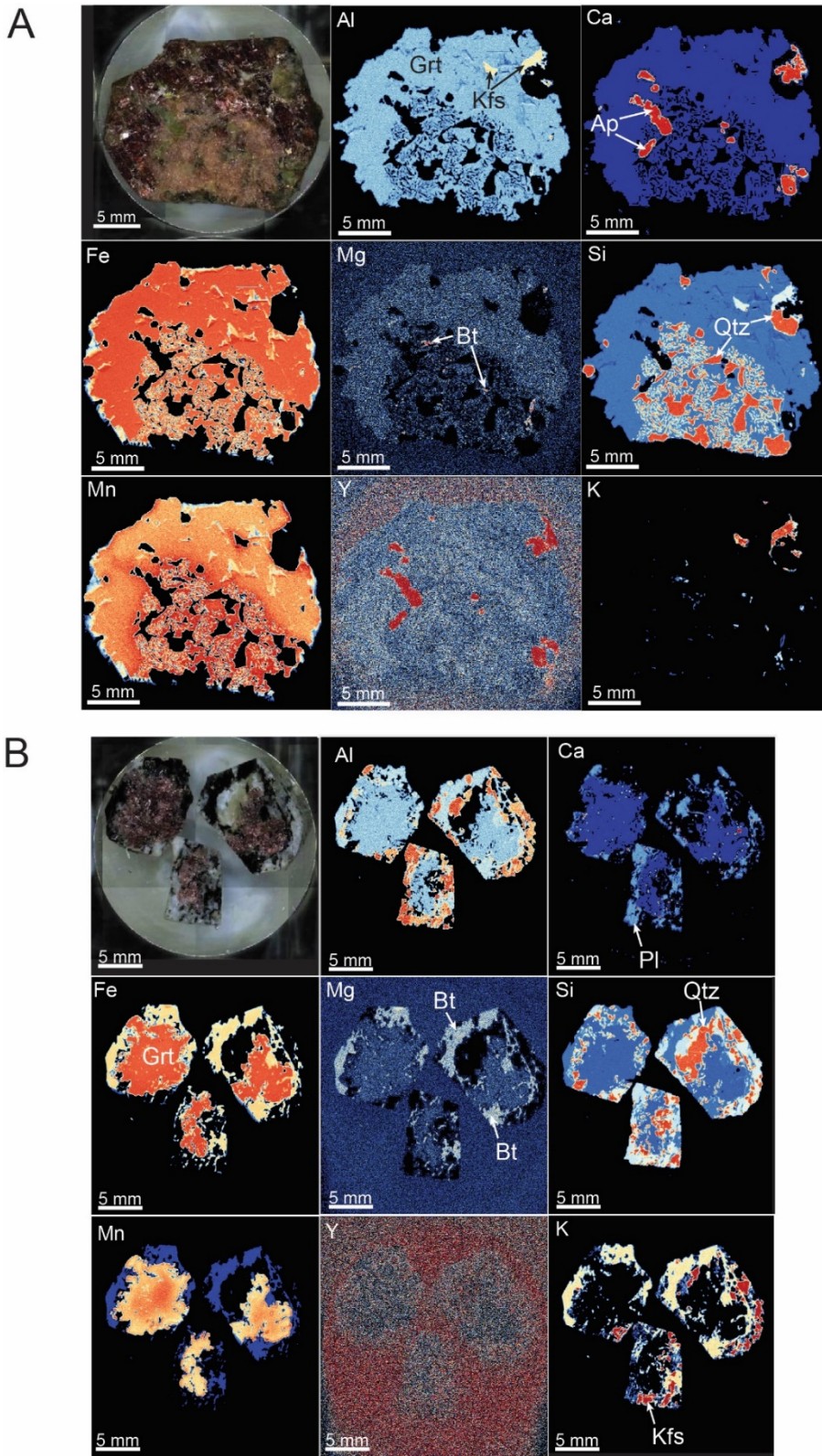

**Figure 5.** A. Garnet from leucosome (MM31) mounted in epoxy with elemental images of the mount obtained by micro-XRF. Images are intensity maps with the colour scale varying from black-blue (low) to red (high). B. Garnet grains from the mesosome (MM30) mounted in epoxy with elemental images of the mount obtained by micro-XRF. Images are intensity maps with the colour scale varying from black-blue (low) to red (high).

### 3.5 Garnet and apatite *in-situ* Lu-Hf geochronology

Two garnet-bearing samples (large garnet within leucosome with apatite inclusions, MM31, and small mesosome garnet, MM30) were analysed using polished epoxy mounts (Fig. 5) for *in situ* Lu-Hf garnet and apatite geochronology at Adelaide Microscopy, University of Adelaide, Australia. Garnet and apatite Lu-Hf dating was conducted over two analytical sessions using a RESOlution-LR 193nm excimer laser ablation system, coupled to an Agilent 8900 ICP-MS/MS. The laser beam diameter was set to 173 μm (garnet) and 120 μm (apatite), and ablation was conducted at 10

Hz repetition rate and a fluence of ~3.5 J/cm$^2$.

The laser-based Lu-Hf method uses a $NH_3$ – He gas mixture in the reaction-cell of the mass spectrometer to promote high-order reaction products of Hf, with a mass-shift of +82, while equivalent Lu and Yb reaction products are minimal (i.e., Hf reacts at a rate of 50-60% while Lu reaction is < 0.003%; Simpson et al., 2021). Consequently, the resulting mass-shifted (+82 amu) reaction products of $^{176+82}$Hf and $^{178+82}$Hf can be measured free from isobaric

interferences. $^{177}$Hf was subsequently calculated from $^{178}$Hf, assuming natural abundances. $^{175}$Lu was measured on-mass as a proxy for $^{176}$Lu (see details in Simpson et al., 2021, 2023). In addition to Lu and Hf isotopes, other trace elements including a selection of other Rare Earth elements (REEs) (details in Supplementary Tab. 5) were measured simultaneously to monitor for inclusions and to characterise the nature of the fluids. However, not every REE was measured as this would compromise the dwell times on the Hf isotopes required for age calculations. For both garnet

and apatite Lu-Hf analysis, NIST 610 was used as a primary reference material (Nebel et al., 2009). For garnet, the reference material Högsbo garnet (reference age 1029 ± 1.7 Ma) was analysed repeatedly to correct for matrix-dependent fractionation (Simpson et al., 2021; Glorie et al., 2024b) and secondary garnet reference material BP-1 (Black Point, South Australia; Glorie et al., 2024b) was used to validate the accuracy of the Lu-Hf dates. BP-1 produced a garnet Lu-Hf isochron age of 1749 ± 15 Ma (Supplementary Tab. 4), which is consistent with the published

monazite U-Pb age of 1745 ± 14 Ma (Lane, 2011).

For apatite, the reference material OD-306 was analysed repeatedly to correct for matrix-dependent fractionation (1597 ± 7 Ma; Thompson et al., 2016). The secondary reference apatites Bamble-1 (Bamble sector, SE Norway; Lu-Hf age of 1102 ± 5 Ma; Glorie et al., 2024a) and HR-1 (Harts Range, NT Australia; Lu-Hf age: 343 ± 2 Ma; Glorie et al., 2022) were used to monitor accuracy. During this study the age obtained for Bamble-1 was 1084 ±

18 Ma and for HR-1 was 344 ± 3 Ma (Supplementary Tab. 5). Apatite U-Pb and trace element analysis was conducted on the same instrumentation as for the Lu-Hf analyses, using identical analytical parameters as in Gillespie et al., (2018) and Glorie et al., (2019), including a laser diameter of 30μm and repetition rate of 5Hz. The primary reference material used was MAD (ID-TIMS U-Pb age 473.5 ± 0.7 Ma; Thomson et al., 2012; Chew et al., 2014). The 401 apatite was used as a secondary standard, producing a weighted mean $^{206}$Pb/$^{238}$U age of 529 ± 2 Ma (Supplementary

Tab. 6). This is in good agreement with the published age (ID-MC-ICP-MS U-Pb age 530.3 ± 1.5 Ma; Thompson et al., 2016).

Isotope ratios and trace element concentrations were calculated in LADR (Norris and Danyushevsky, 2018). Lu-Hf ages were calculated as inverse isochrons using IsoplotR (Vermeesch, 2018; Li and Vermeesch, 2021) with the $^{176}$Lu decay constant of Söderlund et al. (2004) 0.0001867 ± 0.00000008 Ma$^{-1}$. For samples that produced exclusively

high-radiogenic $^{177}$Hf/$^{176}$Hf ratios (< ~0.1), the isochron was anchored to an initial $^{177}$Hf/$^{176}$Hf composition of 3.55 ± 0.05, this value spans the entire range of initial $^{177}$Hf/$^{176}$Hf ratios of the terrestrial reservoir (e.g. Spencer et al., 2020). All presented ages have absolute errors at the 2SD level.

### 3.6 Pressure-temperature pseudosection modelling

Pressure-temperature pseudosections were calculated for sample MM30A using the software package

Theriak/Domino (Capitani and Petrakakis, 2010) and the database of Holland and Powell (2011) for the geologically realistic system MnNCKFMASH (MnO-Na$_2$O-CaO-K$_2$O-FeO-MgO-Al$_2$O$_3$-SiO$_2$-H$_2$O). The 'metapelite set' of models from White et al., (2014a), converted to Theriak-Domino format by Doug Tinkham (see Jørgensen et al., 2019) were applied. These are White et al., (2014b) for orthopyroxene, garnet, biotite, staurolite, chloritoid, cordierite and chlorite; White et al., (2014a) for muscovite and silicate melt; Holland and Powell (2011) for epidote; Holland and

Powell, (2003) for plagioclase; quartz, $H_2O$, kyanite, sillimanite and andalusite are also included as pure phases. Due to the large amount of Mn present in the garnet, MnO was included in the system. However, the low Ti content and absence of Ti bearing minerals makes the inclusion of $TiO_2$ unnecessary. Additionally, a lack of $Fe^{3+}$ bearing phases such as magnetite and the low indicated $Fe^{3+}$ contents in recalculated garnet analysis (see Supplementary Tab. 2) indicated that including ferric iron in the modelling was unnecessary.

265        The presence of leucosomes, fine-grained domains and cuspate grain boundaries in sample MM30A suggest that melt was part of the peak assemblage of the samples. Since it is impossible to know if this melt was retained in the system, a T-$X_{H2O}$ diagram was calculated to indicate an appropriate $H_2O$ value for the P-T diagram (Supplementary Fig. S1). Many thin sections contain a significant amount of apatite, which is also observed as inclusions within garnet (Fig. 5A). Since apatite contains appreciable amounts of CaO, a $T$-$X_{CaO}$ diagram was also generated using the measured

amount of $P_2O_5$ to determine the maximum amount of CaO that could be attributed to apatite (Supplementary Fig. S2).

## 4    Results
### 4.1  Whole-rock geochemistry

To further support this research, 117 whole-rock intermediate and felsic geochemical analyses of the Olkiluoto site
TGG were compiled from Kärki and Paulamäki (2006) including metatexites and diatextites, as well as injected pegmatitic granites and leucosome veins (Supplementary Tab. 7). The diabase dykes from the Olkiluoto region were excluded from the dataset, due that the dykes have intruded into the bedrock considerably later then the main part of the bedrock was formed. Figure 6, shows the main whole-rock compositional characteristics of the Olkiluoto site rocks, distinguishing between typical TGGs and high-P TGGs. Geochemical discrimination diagrams such as TAS
(Fig. 6A), show the difference between normal dominant TGGs ($SiO_2$ 49.60-77.83 wt. % and $P_2O_5$ 0.10-0.23 wt. %) and high-P TGGs ($SiO_2$ 48.45-67.57 wt. % and $P_2O_5$ 0.31-1.73 wt. %) (Fig. 6). In addition, AFM-diagrams effectively illustrate this distinction with high-P TGGs following the tholeiitic series trend, indicative of an older protolith part of migmatites with mainly lower $SiO_2$ contents, and most of the TGGs following a calc-alkaline series trend typical for arc environments fitting well to Svecofennian orogeny (Fig. 6B).

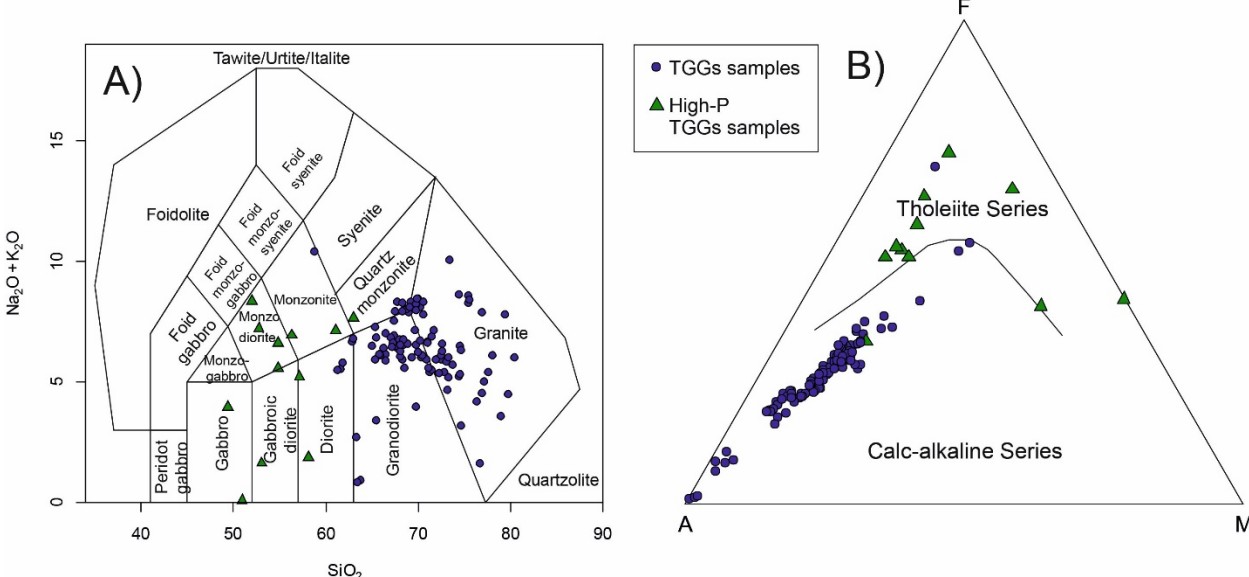

**Figure 6.** The whole-rock geochemistry from Olkiluoto region. A. TAS diagram after Middlemost (1994). B. AFM diagram after Irvine and Baragar (1971). Lithologies: dominant TGG = tonalitic-granitic-granodioritic migmatites including metatexites and diatexites, and high-P TGGs.

## 4.2 Major and trace element mineral chemistry

Garnet major element mineral chemistry for both the large (MM31) and small (MM30) grains are very similar in terms of trends and absolute values (Fig. 7). Garnet from both MM31 and MM30 have significant iron enrichment, with $X_{Alm}$ (=Fe/(Fe+Mn+Mg+Ca)) of 0.77 to 0.80. In the large garnet grains, $X_{Alm}$ is fairly constant across the grain, with slight increases next to quartz inclusions. In the small garnet grains, $X_{Alm}$ is slightly elevated both in the core (0.79) and the rim (up to 0.80; Fig. 7B). $X_{Pyr}$(=Mg/(Fe+Mn+Mg+Ca)) values range between 0.08 to 0.12, with the lowest values found in grain cores (0.08) and directly adjacent quartz inclusions. The highest $X_{Pyr}$ values (0.12) are from the rim, although right at the edge of the grain, the $X_{Pyr}$ content drops abruptly (Fig. 7B, D). The same patterns are observed in both large and small garnet grains (Fig. 7). $X_{Sps}$(=Mn/(Fe+Mn+Mg+Ca)) values vary from 0.11 to 0.08 with the higher values coming from the grain core, directly adjacent to quartz inclusions and at the grain rim (Fig. 7). A similar pattern is observed in both, large and small garnet grains (Fig. 7). $X_{Grs}$ (=Ca/(Fe+Mn+Mg+Ca)) values exhibit flat profiles in both large and small garnet grains, with a constant value of just over 0.02. The micro-XRF maps indicate similar compositional variations, with mostly uniform Fe, Mg and Ca (Fig. 5). The Mn maps show higher values in the core, around quartz inclusions and at the rim of grains. The Y map of the large garnet clearly shows the location of high Y apatite inclusions, which are mostly hosted at the rims of large garnet grains (Fig. 5A). The larger garnet seems to have higher Y contents than the small garnet grains. The rim zone, being inclusion poor, appears to have higher Y than the quartz dominated garnet core however, Y contents collected during Lu-Hf analysis indicate higher values for the grain core (Fig. 5A; Supplementary Tab. 4). The small garnets appear to be uniformly low in Y content (Fig. 5B).

Biotite grains, aligned parallel to the compositional layering, have $X_{Mg}$(=Mg/(Mg+Fe)) values of 0.34 to 0.37, with higher values observed in grains included in garnet. $TiO_2$ content varies from 1.84 to 2.88 wt%. In the large garnet sample, plagioclase exhibits a variable composition with $X_{Ab}$(=Na/(Na+Ca+K)) of 0.76 to 0.96. The sample with small garnet has a more restricted $X_{Ab}$ of 0.74-0.79. K-feldspar is dominantly K-rich with $X_K$(=K/(Ca+Na+K)) of 0.81 to 0.87 in both, the small and large garnet samples.

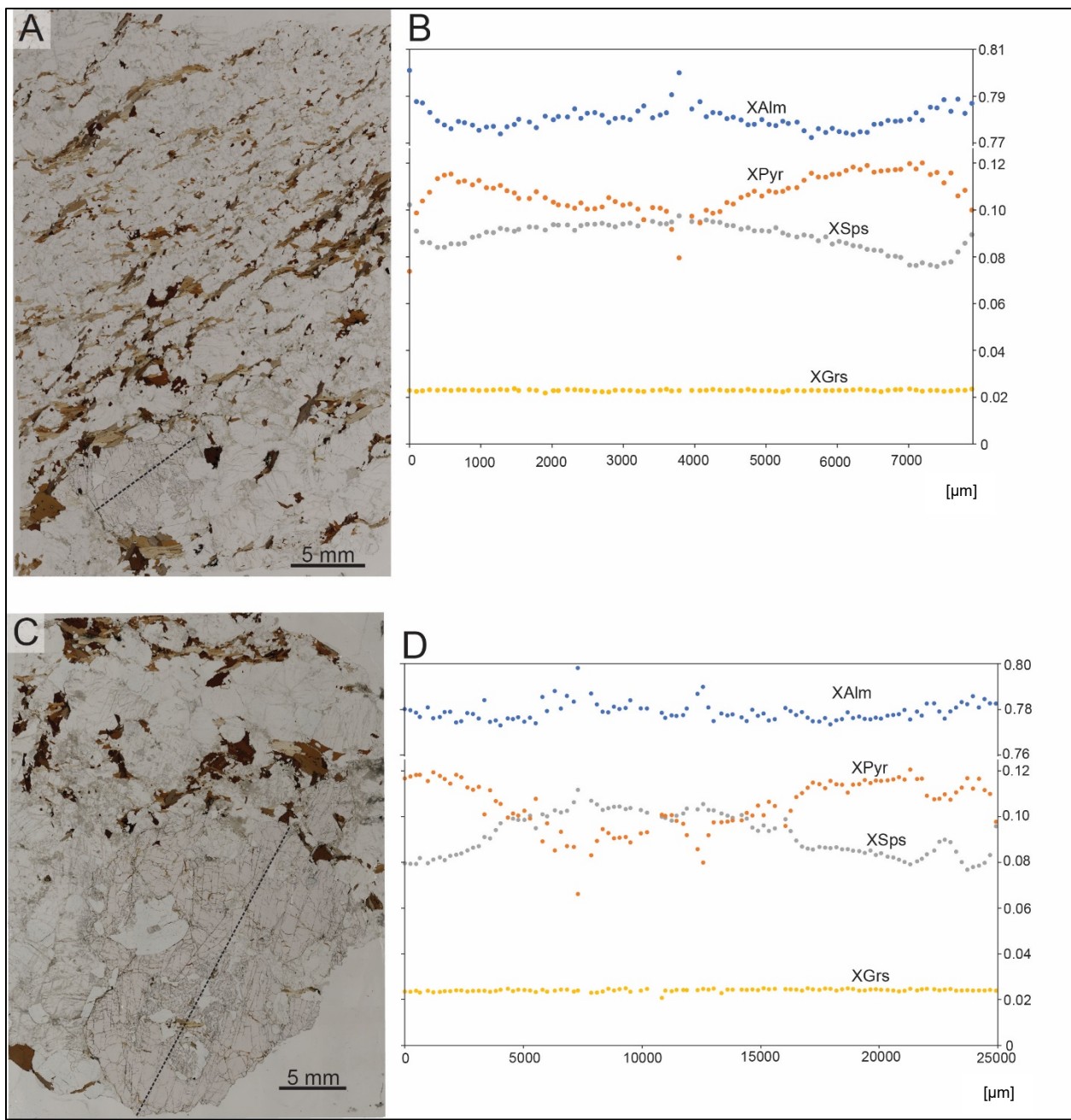

**Figure 7.** Garnet major element zonation. A. Plane polarised image of a thin section of the mesosome domain including the location
of a small garnet grain (MM30) used for the EPMA traverse. The location of the traverse is indicated with the thin dashed line. B.
Garnet traverse for the grain in Fig. 7A. The traverse starts from the left side of the garnet grain. C. Plane polarised image of a thin
section from the leucosome with a large garnet grain (MM31). The location of the traverse is indicated with the thin dashed line.
D. Garnet traverse for the grain in Fig. 7C. The traverse starts from the left side of the garnet grain.

315

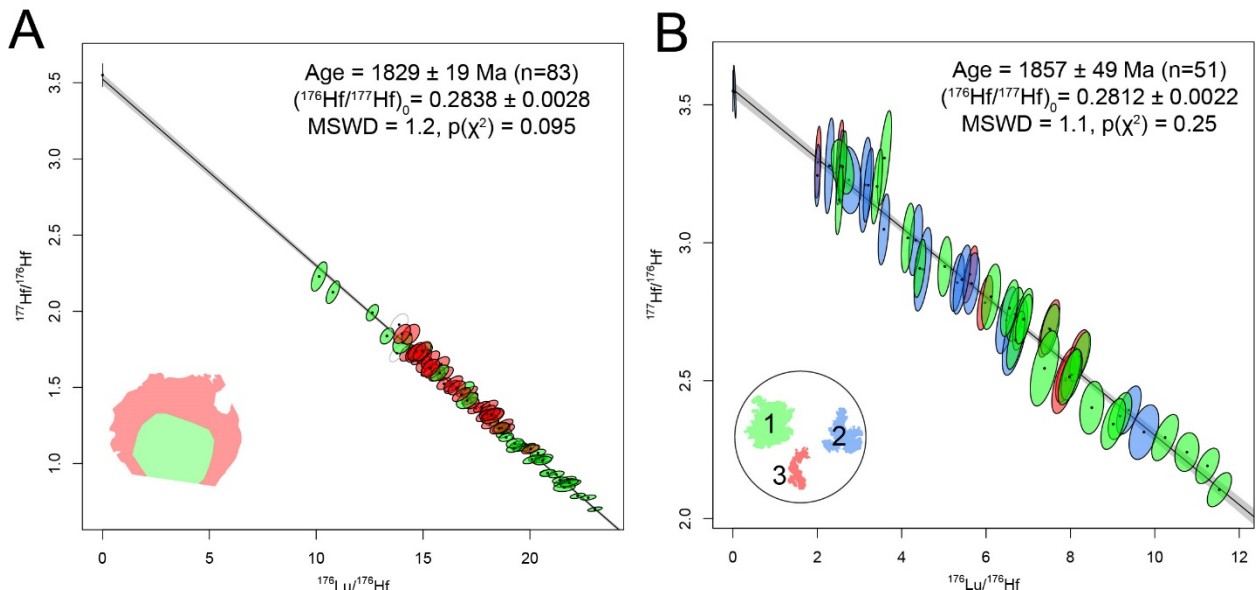

**Figure 8.** A. Lu-Hf inverse isochron for the large garnet sample, MM31. Analyses are coloured based on their location within the garnet grain (green=core, red=rim). B. Lu-Hf inverse isochron for the small garnet sample, MM30. Analyses are coloured based on which garnet grain they were obtained from (see inset for colour key).

## 4.3 Garnet *in-situ* Lu–Hf Geochronology

Two garnet samples (MM30 and MM31) were targeted for *in situ* Lu-Hf geochronology. One large garnet (MM31) hosted in a leucosome and three small grains (MM30) obtained from the mesosome. From the large grain (MM31), a total of 84 analyses were conducted with 42 targeting the grain core and 42 at the grain rim (see Supplementary Fig. S3 for spot locations). Two analyses were excluded from the age calculations due to the presence of inclusions. When all the data are plotted on an isochron anchored to an initial $^{177}Hf/^{176}Hf$ ratio of $3.55 \pm 0.06$ (covering the range of terrestrial values; Mark et al., 2023) the result is an isochron age of $1829 \pm 19$ Ma ($n$=83, MSWD = 1.2; Fig. 8A; one analysis was duplicated with different segments of the signal selected; supplementary Tab. 4). Analyses obtained from the grain core have a larger spread in $^{176}Lu/^{176}Hf$ ratios yielding an age of $1828 \pm 11$ Ma ($n$=43, MSWD = 1.3, one analysis was duplicated with different segments of the signal selected; Supplementary Fig. S4), whereas the measurements from the garnet rim data give an identical isochron age with a larger uncertainty of $1828 \pm 21$ Ma ($n$=40, MSWD= 1.1, two data points were excluded due to inclusions, Supplementary Fig. S5). Garnet cores have Lu contents of 10 to 70 ppm (average is 35 ppm) when calibrated to an internal standard of 12 wt% Al. Garnet rims have Lu contents of 10 to 25 ppm (average is 18 ppm).

The small garnet grains were targeted with 51 analyses in total (see Supplementary Fig. S6 for spot locations) and have a restricted $^{176}Lu/^{176}Hf$ ratio range, resulting in an anchored isochron age of $1857 \pm 49$ Ma ($n$=51; MSWD = 1.1; Fig. 8B). If the grains are plotted separately, they all produce the same age within uncertainty, but due to the smaller number of analyses the errors are larger. The small garnets have Lu contents ranging from 10 ppm to below the detection limit (average is 4 ppm).

## 4.4 Apatite Lu-Hf and U-Pb Geochronology

Four apatite inclusions in the rim of the large garnet (MM31) were dated using both Lu-Hf and U-Pb methods in two separate analytical sessions. The Lu-Hf data are mostly highly radiogenic (38 of 47 analyses with $^{177}Hf/^{176}Hf$ ratios <0.1) and define an anchored Lu-Hf isochron age of $1782 \pm 19$ Ma (MSWD = 1.6; Fig 9A). Alternatively, calculating a weighted mean common-Hf corrected Lu-Hf age (for apatites with $^{177}Hf/^{176}Hf$ ratios <0.1) returns a similar age within uncertainty of $1784 \pm 8$ Ma (MSWD = 1.3, $n$=38). The apatite REE spidergrams indicate slightly enriched light

REEs, a pronounced negative Eu anomaly and flat HREE profiles (Fig. 9C). The investigated apatite compositions plot in the HM field of the apatite classification biplot of (O'Sullivan et al., 2020), indicating apatite crystallized during partial high-grade metamorphic melting processes (Fig. 9D). The apatite U-Pb data plot on a linear trend with some slight scatter (Fig. 9B). An isochron based on 40 out of 45 analyses produces a lower intercept age of $1778 \pm 16$ Ma (MSWD = 0.41). The 5 analyses excluded from this isochron could be related to partial inheritance from an older event or isotopic disturbance (U-loss; Fig. 9B).

## 4.5 Pressure-temperature pseudosection modelling

For sample MM30A, the $T$-$X_{H2O}$ diagram (Supplementary material Fig. S5) indicates that the interpreted peak assemblage of garnet + plagioclase + K-feldspar + biotite + quartz + melt is present at $H_2O$ contents >0.25 on the binary diagram. For this reason, $H_2O$ was set at 0.3 for further calculations corresponding to a $H_2O$ content of less than 1 wt%. A $T$-$X_{CaO}$ diagram was calculated to investigate the impact of apatite on the whole rock CaO content. The diagram was created using the whole rock composition on one side, with the other side indicating a reduction in CaO using the $P_2O_5$ content of the whole rock to indicate the maximum amount of CaO attributable to apatite. In the $T$-$X_{CaO}$ diagram, the interpreted peak assemblage occurs only on the right side of the diagram suggesting that only a modest amount of CaO (corresponding to a 25% reduction) needs to be removed to account for apatite in the sample. Thus, the P-T diagram for sample MM30A was calculated at 0.75 of the $T$-$X_{CaO}$ diagram (indicating that of a total 100% CaO that could be attributed to apatite, only 25% was removed; see Supplementary Fig. S2).

The P-T diagram for sample MM30A has the interpreted peak assemblage field of garnet + plagioclase + K-feldspar + biotite + quartz + melt present over a large range of pressures and temperatures, extending from 2.5 kbar to over 10 kbar and from 650 °C to 800 °C (Fig. 10). The garnet compositional isopleths which correspond to the garnet compositional range observed in both the large and small garnet grains ($X_{Alm}$: 0.8-0.78; $X_{Pyr}$: 0.12-0.1; $X_{Sps}$: 0.1-0.08; $X_{Grs}$<0.04; Fig. 7) occur in the lower pressure part of the field with compositional overlap occurring in the range of 3-5 kbar and ca. 700 °C (Fig. 10).

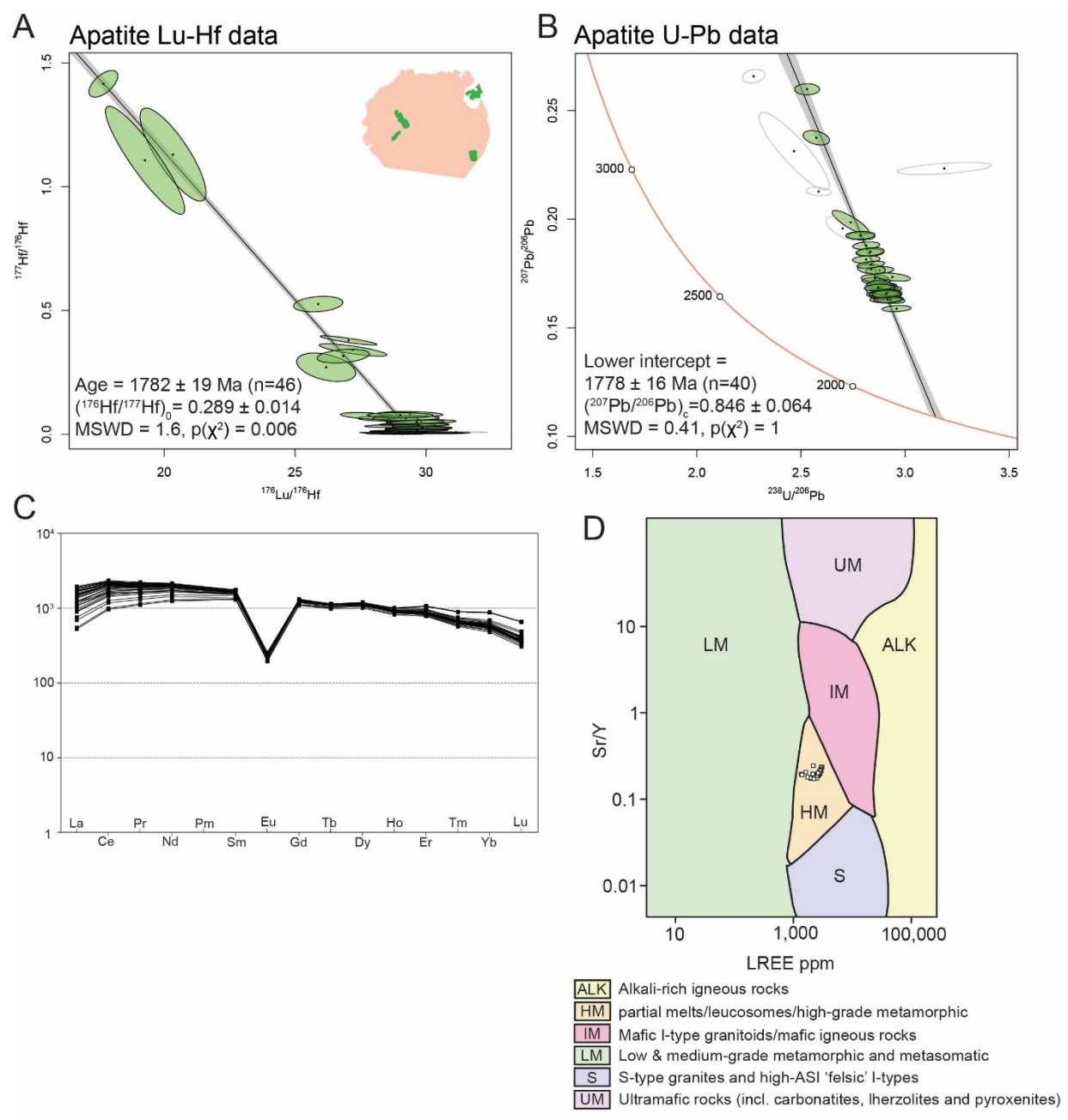

**Figure 9.** A. Lu-Hf inverse isochron for apatite occurring within the large garnet grain (green grains in the inset image). B. Apatite U-Pb data plotted on a Terra-Wasserburg concordia plot. C. Apatite REE spidergram normalised to chondrite (McDonough and Sun, 1995). D. Apatite classification biplot of O'Sullivan et al., (2020) based on Sr/Y vs ΣLREE (La-Nd). The apatite analysed in this study is plotted on this diagram as small white squares and plots exclusively in the HM field.


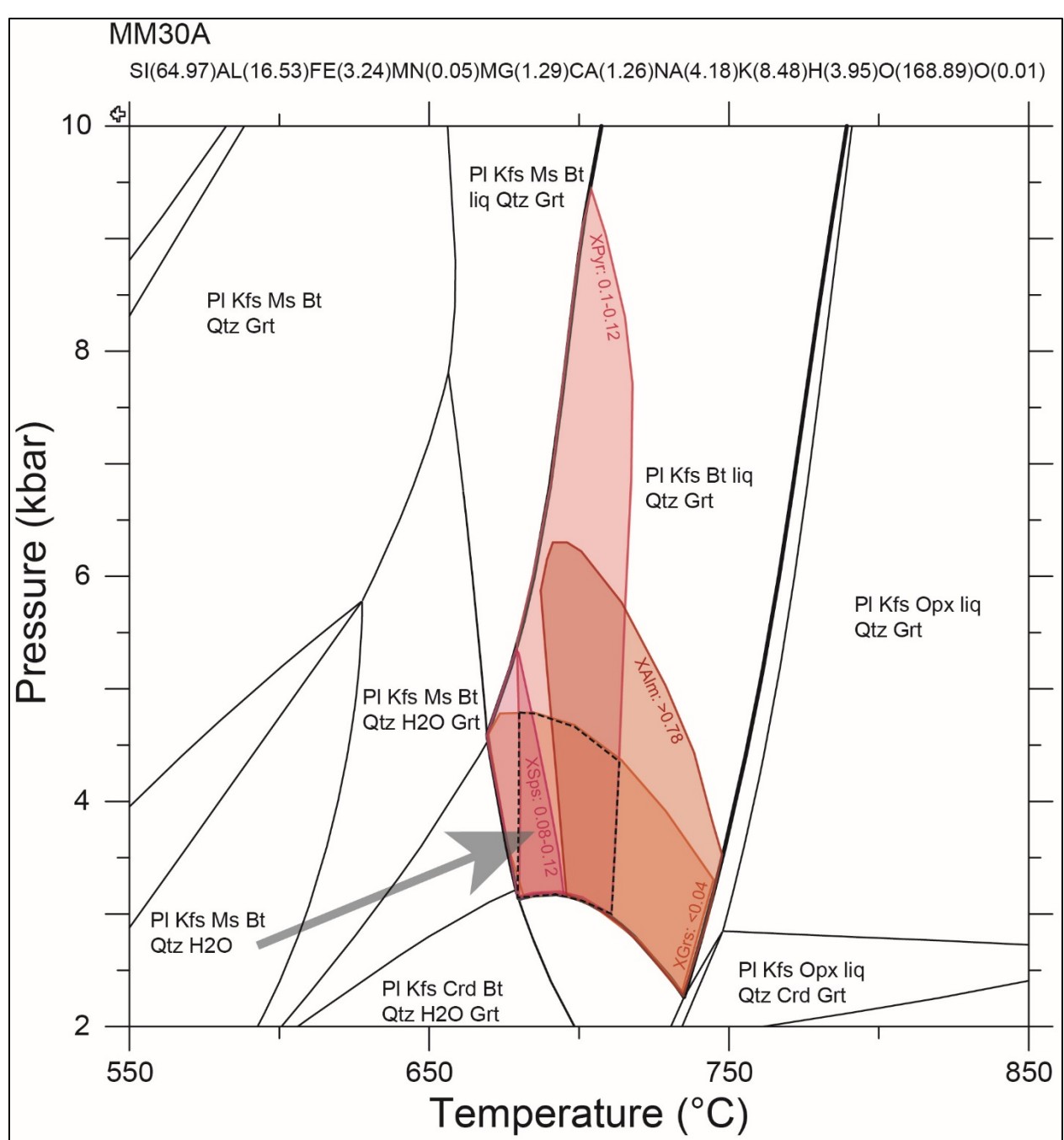

**Figure 10.** P-T pseudosection for sample MM30A using the bulk composition shown at the top of the diagram (see Supplementary Tab. S1). The T-$X_{H2O}$ and T-$X_{CaO}$ diagrams used to investigate this composition are given as Fig. S5 and S6. The shaded red areas indicate regions matching the garnet composition observed in the sample ($X_{Alm}$: 0.8-0.78; $X_{Pyr}$: 0.12-0.1; $X_{Sps}$: 0.1-0.08; $X_{Grs}$<0.04). The black dashed box is the field of overlap in the P-T pseudosection and the grey arrow indicates the direction of increasing garnet mode (see Supplementary Fig. S7).

## 5    Discussion
### 5.1   Significance of the age data and metamorphic constraints

The small garnet (MM30) Lu-Hf data produces an age of 1857 ± 49 Ma with a large uncertainty as a result of the low Lu contents in the small garnet. The analyses from the large garnet (MM31) produce a Lu-Hf age of 1829 ± 19 Ma. These results are statistically indistinguishable, meaning that if there is a difference in age between the small and large garnets, it is not possible to define this with the Lu-Hf method. To distinguish these populations, more data is required targeting high Lu domains within the small mesosome garnets or obtaining additional samples. Texturally, the small
garnets occur in the rock mesosome, while the larger garnets occur in the leucosome. Plausibly, this may mean that the smaller garnets are older. However, both the sample mesosomes and leucosomes are deformed by the same foliation (Figs. 3, 4, 7) suggesting it is equally feasible that all garnet grew in response to a single phase of deformation. The mesosome garnets may be small due to having a limited equilibrium volume in comparison with the leucosome garnet which formed in melt. This may also be responsible for the variable Lu contents. The small mesosome garnets
have the lowest Lu contents (average of 4 ppm), potentially due to a small equilibrium volume. The leucosome garnets growing in melt may have had a larger equilibrium volume due to more effective transport of elements within the melt resulting in the highest Lu contents in the grain cores (average of 38 ppm) with lower values in the rim (average of 21 ppm) following garnet (and melt) crystallization. An additional support for this scenario is the largely flat major element zonation of both large and small garnets with similar absolute values. This would indicate that both small and
large garnets equilibrated at the same elevated T conditions. Thus, based on our current results we interpret that the large garnets and possibly also the small garnets grew in response to one metamorphic event at 1829 ± 19 Ma (MSWD of 1.3) using the age obtained from the large leucosome garnet (MM31). Recently, Smit et al., (2024) showed that even at high-grade conditions (> 800 °C), REE diffuse slowly in natural garnet making Lu-Hf chronology extremely robust. Despite a long and complex history of upper amphibolite facies metamorphism and magmatism in the region
(Väisänen and Hölttä, 1999; Hölttä and Heilimo, 2017; Torvela and Kurhila, 2020), it is likely that the Lu-Hf results reflect garnet growth ages.

        The Lu-Hf and U-Pb ages obtained for apatite grains hosted inside large garnets are similar within error, producing ages of 1782 ± 19 Ma and 1778 ± 16 Ma respectively. The Lu-Hf system in apatite is considered to have a higher closure temperatures (~670-730 °C; Glorie et al., 2024a) than U-Pb in apatite (350-570 °C; Chew and Spikings,
2021). Thus, a similarity in the U-Pb and Lu-Hf ages is indicative of fast cooling at the time of apatite growth. The difference in age between the apatite and the hosting garnet grain (large garnet gave an isochron age of 1829 ± 19 Ma) could indicate that the apatite grains are in contact with the mesosome and the symplectitic quartz that occurs in the grain core (thus open to grain boundary fluid conduits which may have resulted in resetting at ca. 1780 Ma) or alternatively the sample may have stayed at an elevated temperature (> 700 °C) from ca. 1830 to 1780 Ma. This would
be consistent with the relatively flat major element compositional profiles in garnet. Producing nearly flat profiles in a large garnet grain would require a prolonged period (likely on the order of 10s of millions years) at temperatures above 650 °C  (Caddick et al., 2010).

        Mänttäri et al., (2006) obtained U-Pb zircon and monazite ages from the same TGG gneiss unit sampled in this study (sample site 2006 in Fig. 2). The zircon U-Pb data indicate a crystallization age of ca. 1860 Ma
for the tonalite with Archean (ca. 2.7 Ga) and Paleoproterozoic (2.0-1.9 Ga) inheritance. Pegmatitic granite dykes, interpreted to be related to a migmatization (partial melting) event, yield zircon ages of ca. 1830-1790 Ma with interpreted inheritance of Archean (ca. 2.7 Ga) and ca. 1865 Ma. A monazite U-Pb age from a pegmatite of 1823 ± 3 Ma is interpreted as a minimum age for pegmatitic dykes (Mänttäri et al., 2006), these are ages consistent with the dates obtained in this study and interpreted as ages for the high-temperature partial melting event.
The P-T diagram for sample MM30A indicates peak P-T conditions at 3-5 kbar and ~700 °C. This result is consistent with previous studies in the Olkiluoto site by Tuisku and Kärki (2010), defining that migmatisation of pelitic gneisses due to the dehydration melting of biotite, sillimanite, plagioclase and quartz in the temperature between 660 to 700 °C and pressure of 3.5-4 kbar.  The arrow in Figure 10 indicates a proposed prograde P-T path defined by the increase in garnet mode. The P-T path is also parallel to grossular compositional isopleths, producing flat

compositional zonation in $X_{Grs}$ along this path, as well as an increase in $X_{Pyr}$ and decrease in $X_{Sps}$ consistent with the compositional zonation observed in the sample.

Based on the interpretation above the garnet ages likely represent growth ages, representing a period of prograde evolution that occurred at 1829 ± 19 Ma. The large apatite grains that occur as inclusions at the rim of the large garnet preserve Lu-Hf and U-Pb ages of 1782 ± 19 Ma and 1778 ± 16 Ma. As discussed above, this presents two options for the metamorphic evolution of the sample. Either the sample cooled from peak conditions at ca. 1830 Ma and was then reheated to nearly identical conditions at ca. 1780 Ma or high temperature conditions persisted from 1830 to 1780 Ma at which point, rapid cooling from 700 to 400-500 °C (closure T of U-Pb in apatite) occurred. The latter metamorphic evolution would be consistent with a regional high temperature, low pressure event that in other areas has been interpreted to have persisted for this period (Hölttä and Heilimo, 2017). Potentially these conditions persisted for ca. 50 Ma to ca. 1780 Ma, where the transition from ductile to brittle tectonics has been inferred by Torvela et al. (2008). The study by Nordbäck et al. (2022) from the Olkiluoto site, concluded that the ductile to brittle transition were related to the late Svecofennian tectonics where structurally controlled accumulation and percolation of metamorphic fluids occurred in ductile shear zones that finally triggered the first embrittlement in Olkiluoto and emplacement of quartz veins in faults.

## 5.2 The role of Olkiluoto region in the tectonic setting of S Finland

The P-T-t conditions of Olkiluoto are summarized in Figure 11, and a comparison to surrounding Svecofennian tectonic belts is shown. The Tampere Belt experienced greenschist (central Tampere Belt) - to amphibolite (east and west Tampere Belt) - facies conditions with peak metamorphism interpreted to occur at ca. 1.88 Ga (Mouri et al., 1999). Lahtinen et al., (2017) also obtained a garnet age from a mica schist in the Tampere Belt producing an age of 1.81 Ga, which they suggest indicates that elevated temperature conditions continued until this time. The Pirkanmaa Belt contains upper amphibolite- to granulite-facies migmatitic rocks, that experienced peak metamorphic conditions at 4-5 kbar and 750-700 °C at ca. 1.88 Ga (Mouri et al., 1999). There is also evidence of younger monazite (ca. 1.85 Ga, Hölttä et al., 2020) and a range of garnet Sm-Nd ages from 1.89-1.84 Ga (Lahtinen et al., 2017; Mouri et al., 1999). The Häme Belt metapelitic rocks are either sillimanite-muscovite gneisses or schists, whose Al-rich layers often have andalusite, staurolite or even cordierite porphyroblasts. The estimated P-T conditions on the basis of these stable mineral assemblages, is inferred during the peak metamorphism to be 3-4 kbar and 530-580 °C (Hölttä and Heilimo, 2017). The age of peak metamorphism is poorly defined but is generally interpreted to have occurred at 1.83-1.80 Ga, based on U-Pb zircon and titanite ages in altered diorite (Saalmann et al., 2010; Hölttä et al., 2020). Saalmann et al. (2009) indicate that the Häme Belt experienced an earlier compressional event at ca. 1.88-1.86 Ga, by subduction followed by an extensional event caused by slab rollback event. This is supported by the study from Lahtinen et al., (2017), that found 1.90-1.86 Ga metamorphic overgrowth in zircons from a metapsammitic rock. The southernmost Uusimaa Belt is proposed to have experienced crustal extension at 1.86-1.84 Ga, followed by a transpressional event producing granulite facies peak conditions of 4-7 kbar and 750-825 °C at 1.83-1.80 Ga (Väisänen and Hölttä, 1999; Mouri et al., 2005; Skyttä and Mänttäri, 2008).

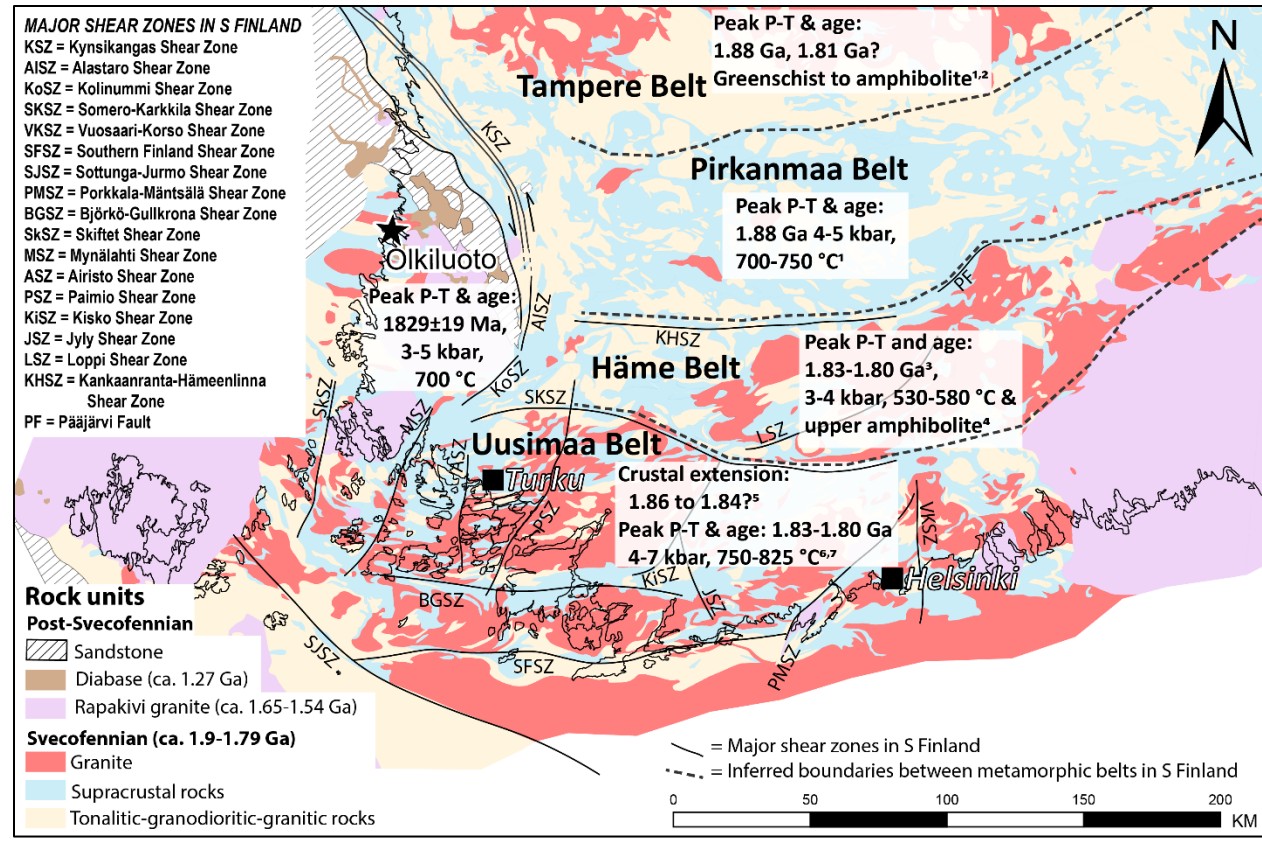

**Figure 11.** The geological map of S Finland with the different tectonic and metamorphic belts and significant shear zones. Bedrock map scale 1:200 000, from Geological Survey of Finland (Geological Survey of Finland, 2022). Shear zones adapted from (Heeremans et al., 1996; Pitkälä et al., 2018; Reimers et al., 2018; Torvela et al., 2008; Torvela and Kurhila, 2022; Väisänen et al., 2014; Väisänen and Skyttä, 2007); Lahtinen et al., (2023). The peak P-T conditions and metamorphic ages are derived from the following references: 1) Mouri et al. (1999), 2) Lahtinen et al. (2017), 3) Hölttä et al. (2020), 4) Hölttä and Heilimo (2017), 5) Skyttä and Mänttäri (2008), 6) Mouri et al. (2005) and 7) Väisänen and Hölttä (1999).

The Olkiluoto region and the Häme Belt (situated ca. 75 km southeast from Olkiluoto; Fig. 11) have similar crystallisation ages (Kähkönen, 2005; Mänttäri et al., 2006) as well as metamorphic and tectonic histories during the ca. 1.88-1.79 Ga Svecofennian orogeny. Previous metamorphic studies in the Häme Belt have shown that that it contains supracrustal rocks that have been interpreted to have two metamorphic peaks: first in amphibolite facies conditions at ca. 1.88-1.86 Ga (Nironen, 1999; Väisänen et al., 2002; Kähkönen, 2005; Kara et al., 2021) and the latter during a high-T event that peaked at ca. 1.83-1.81 Ga (Väisänen et al., 2002). Kurhila et al. (2011) defined that Southern Finland was subjected to a prolonged anatexis event and latter event with the emplacement of late-orogenic leucogranites, Torvela and Kurhila (2022) defined that in addition to a hot anatexis, a late-orogenic event was coupled to migmatitization and formation of major shear zones in a transpressional tectonic regime (Fig. 11). Our novel garnet and apatite Lu-Hf age data together with apatite U-Pb age data, that is coupled with pseudosection modelling and supporting data from the previous study on the structural setting by Engström et al. (2022), supports the same evolution for the Olkiluoto region and the Häme belt. Thus, the age indicated by the Lu-Hf geochronology (1829 ± 19 Ma) of the garnets from this study are interpreted to show the latter distinct metamorphic peak (M2) for the Olkiluoto site. The later metamorphic peak (M2) caused migmatisation of the Olkiluoto site's protolith tholeiitic TGG rocks, which have a high $P_2O_5$ content, resulting in the formation of dominant calc-alkaline series TGG rocks via anatexis during this period (see Chapter 4.1). Additionally, the structural study by Engström et al. (2022), that defines that later formed pegmatitic leucosomes (M2) crosscut earlier compositional banding, implying that they are younger, and the

metamorphic evolution in this study, infers similar metamorphic and tectonic signatures with other studies from the Häme Belt (Kähkönen, 2005; Hölttä and Heilimo, 2017; Kara et al., 2021; Lahtinen et al., 2023).

The study by Saalmann et al. (2009) defined that the observed metamorphic peak in the Häme Belt was followed by a hydrothermal event with formation of shear zones and mineralization. Even though no mineralization indicative of a hydrothermal event is observed in the Olkiluoto site, a similar hydrothermal and shearing-induced event is exemplified by the formation of a certain type of diatexitic migmatite with roundish quartz-feldspar megacrysts (Engström et al., 2022), which are interpreted to be related to the prolonged cooling after the peak P-T conditions of 3-5 kbar and around 700 °C (this study). The ductile-brittle transition in the Olkiluoto site is interpreted to be related to metamorphic fluids and the roundish quartz-feldspar megacrysts is defined to have a tectonic connection to the first brittle tectonic faulting event in Olkiluoto (Engström et al., 2022; Nordbäck et al., 2022). Thus, it is plausible that latest tectonic hydrothermal and shearing-induced events in Häme Belt and the Olkiluoto site can be syn-tectonic.

At the same time, the Uusimaa Belt (Fig. 11) mainly underwent granulite-facies metamorphism at ca. 1.84-1.81 Ga (Väisänen et al., 2002; Mouri et al., 2005; Skyttä and Mänttäri, 2008) inferring slightly deeper crustal depth than the Häme Belt. These belts and the Olkiluoto site, seem to have a similar geothermal gradient, where the Uusimaa Belt, Häme Belt and Olkiluoto site have geothermal gradients of 63 °C/km, 57°C/km and 64°C/km respectively (based on P-T estimates and crustal density of 2.75 g/cm$^3$). The interpretation further south in the Uusimaa Belt, where shear zones and the anatectic melting are strongly coupled to each other (Torvela and Kurhila, 2020; Lahtinen et al., 2023), indicates that the whole of the southern Finland domain was subjected to a long hot orogenic evolution with several crustal-scale melt pulses, as described by Chardon et al., (2009) for Precambrian accretionary orogens.

**5.3 Implications for southern Finland tectonic framework**

The tectonic framework for southern Finland possibly represents the distal regions in a back-arc basin complex that formed above the retreated subduction zone in the west, and slab rollback caused (oblique) extension in the upper plate and asthenospheric upwelling in back-arc regions, following the proposed models (Collins, 2002; Hermansson et al., 2008; Saalmann et al., 2009; Kara et al., 2021). This caused high heat flow and decompression melting and mafic underplating of the thinned continental crust, giving rise to melt production and intense magmatic activity as well as granulite facies metamorphism in its deeper parts (Väisänen and Hölttä, 1999). This infers different crustal depths for the Häme Belt and Uusimaa Belt (Saalmann et al., 2009; Torvela and Kurhila, 2020). The presence of several shear zones in southern Finland is indicative of a transpressive tectonic regime that could be explained by strain partitioning in the oblique tectonic regime where contractional segments were coupled to folding and thrusting, which is observed especially in the Häme Belt (Kara et al., 2021; Nironen, 1999; Pitkälä et al., 2018; Saalmann et al., 2009). While, in the Uusimaa Belt, the tectonic regime exhibits more transtensional shear zones (Väisänen and Skyttä, 2007; Torvela and Kurhila, 2022).

The Olkiluoto region is inferred to be coupled to the Häme Belt (see Chapter 5.2) with the Somero-Karkkila Shear Zone (SKSZ) acting as a metamorphic boundary between the Häme Belt and the Uusimaa Belt. However, the mosaic structure with shear zones and different crustal blocks that is characteristic for southern Finland show that detailed comprehensive studies coupling to both, structural geology and metamorphic studies is essential when determining the character and tectonic evolution of the crystalline bedrock in a high-grade environment that is prevailing in this tectonic domain. The impact of these crustal scale shear zones in the tectonic framework of southern Finland is evident, but the kinematic constraints and age relationships are still poorly understood. However, most of these shear zones show ductile deformation signatures inferring deeper and hotter origin, indicating formation before the ductile-brittle transition interpreted at ca. 1.78 Ga (Nordbäck et al., 2024).

**5.4 Connection to Ljusdal lithotectonic unit**

This study is an additional piece of the puzzle linking the crustal units of southwest Finland and central Sweden. Only a limited number of studies are available, and clearly more studies are needed especially in the Bothnian Basin separating Sweden and Finland to unravel if e.g., a failed rifting event is the cause of the separation and why

sedimentary units are mainly present offshore (Korja and Heikkinen, 2005; Buntin et al., 2019; Fig 1). Structural studies from southwestern Finland by Nordbäck et al. (2024) emphasize that large N-S structures are related to this rifting and to development of Mesoproterozoic sedimentary basin at the centre of Fennoscandian shield, located beneath the Bothnian Sea (e.g. Kohonen and Rämö, 2005). Thus, it is reasonable to assume that prior to this rifting event central Sweden and southwestern Finland were connected as proposed by Engström et al. (2022) and by Luth et
al. (2024) in the lithotectonic map of Fennoscandia. Further evidence is supported by the presence of similar mineralogy cordierite–sillimanite–garnet mineral assemblage, in both Ljusdal and Olkiluoto (Högdahl and Bergman, 2020; Engström et al., 2022). The prolonged ductile deformation with several crustal-scale melt pulses indicates that the Olkiluoto site is similar to the Ljusdal lithotectonic unit in central Sweden (Högdahl et al., 2012; Engström et al., 2022; Fig. 1). Earlier studies by Engström et al., (2022) and Saukko et al., (2020) defined that the area was subjected
to a high-grade migmatitic event during an approximate time span of 90 Ma between 1.87–1.78 Ga, with two main migmatite-producing events. Hence it is plausible, that the Häme Belt, including the Olkiluoto site, is connected to the Ljusdal lithotectonic unit. The major crustal-scale Kynsikangas shear zone located 40 km northeast of Olkiluoto possible possibly marks the tectonic boundary between the lithotectonic units (Fig. 11; Engström et al., 2022). The Saimaa orocline is deduced as a big suture zone between WFS and SFS in studies in eastern Finland (Lahtinen et al.,
2022) and a likely continuation of that is the Kynsikangas shear zone with a possible continuation to the Hassela shear zone in central Sweden (Reimers et al., 2018; Högdahl and Bergman, 2020; Lahtinen et al., 2023).

## 6    Conclusions

The previously published structural data (Engström et al., 2022) together with the metamorphic data presented in this
paper, suggest that the Häme Belt including the Olkiluoto area and the Ljusdal lithotectonic unit, share a similar deformation history and metamorphic P-T-t conditions (Högdahl et al., 2008; Hölttä and Heilimo, 2017; Högdahl and Bergman, 2020; Saukko et al., 2020; Lahtinen et al., 2023; Luth et al., 2024). Both areas show a younger ca. 1.83 Ga amphibolite-facies metamorphic peak and possibly an older ca. 1.86 Ga metamorphic event.

The main outcome from this study are as follows:
- The garnets in the TGG type rock at the Olkiluoto site were studied and *in situ* Lu-Hf geochronology defined a metamorphic peak at $1829 \pm 19$ Ma.
- The P-T modelling at the site indicates peak P-T conditions of 3-5 kbar and around 700 °C.
- The metamorphic evolution in southern Finland is poorly constrained due to complex structural
geological evolution. This study provides valuable input to better constrain the metamorphic similarities between the Olkiluoto site and the Häme Belt. However, more metamorphic studies in both the Uusimaa Belt and the Häme Belt are needed to further constrain this.
- The Olkiluoto site is well located between southern Finland and central Sweden and thus represents a key location to define a missing link between the Swedish and Finnish lithotectonic units. This study
presents new results coupling these crustal units together.

**Credit authorship contribution statement:**

**Jon Engström:** Conceptualization, Methodology, Interpretation, Visualization, Writing - original draft **Kathryn Cutts**: Conceptualization, Methodology, Interpretation, Visualization, Writing - review & editing. **Stijn Glorie**: *In*
*situ* Lu-Hf geochronology, Validation, Writing - review. **Esa Heilimo:** Whole-rock geochemistry, Writing - review & editing. **Ester M. Jolis:** Micro-XRF analysis, Writing - review. **Radoslaw M. Michallik:** EPMA analysis, Writing - review.

**Competing interests**

The contact author has declared that none of the authors has any competing interests.

**Acknowledgements**

We are grateful to Posiva Oy for the support during the field work, and access to the data. Micro-XRF was supported by the Academy of Finland via RAMI infrastructure project (#337560). The reviews by the officially appointed Journal Reviewers S. Luth and two Anonymous reviewer both helped to improve the manuscript.

**Supplementary data**

1. Supplementary Figures

2. Supplementary Table

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
