# Peer review of "Insights into the tectonic evolution of the Svecofennian orogeny based on *in situ* Lu-Hf dating of garnet and apatite from Olkiluoto, SW Finland"

_EGUsphere, 2024_

## Referee Comment (RC1)

[referee-annotated manuscript omitted]

---

## Referee Comment (RC2)

[revised manuscript text omitted]

**Supplementary Figures**

[Figure]

Fig. S1. T-XH2O diagram at 6000 bar for sample MM30A. The interpreted peak assemblage of garnet-plagioclase-biotite-K-feldspar-quartz-melt is highlighted in bold. The red fields indicate the XSps and XPyr compositions that are present in the garnet grains of the sample. The bold line indicates the H2O value used to calculate the TXCaO and P-T diagrams.

[Figure]

Fig. S2. T-XCaO diagram for sample MM30A at 4000 bar. The interpreted peak assemblage of garnet-plagioclase-biotite-K-feldspar-quartz-melt is highlighted in bold. The red fields indicate the XSps, XPyr and XAlm compositions that are present in the garnet grains of the sample. The bold line indicates the CaO value used to calculate the P-T diagram.

[Figure]

Fig. S3. Lu-Hf spot locations for large garnet.

[Figure]

Fig. S4. Calculated Lu-Hf isochron for analyses obtained from the garnet core only.

[Figure]

Fig. S5. Calculated Lu-Hf garnet isochron using analyses obtained from the rim only.

[Figure]

Fig. S6. Lu-Hf spot locations for small garnet.

---

## Author Response (AR1)

**REVIEW RESPONSES TO MANUSCRIPT VERSION 1**

Please find our manuscript titled "***Insights into the tectonic evolution of the Svecofennian orogeny based on in situ Lu-Hf dating of garnet and apatite from Olkiluoto, SW Finland***" which we submit for consideration to be published in the Solid Earth.

Sincerely on behalf of all authors,

Jon Engström

**Reviewer #1**

Comments by reviewer Stefan Luth

*Title:* ***Insights into the tectonic evolution of the Svecofennian orogeny based on in situ Lu-Hf dating of garnet from Olkiluoto, SW Finland***

*Author(s): Jon Engström et al.*

*MS No.: egusphere-2024-2034*

*MS type: Research article*

**General comments**

As the title clearly reflects, the manuscript addresses a relevant question on the tectonic evolution of the Svecofennian orogeny by applying a relatively novel approach: In situ dating of garnet and apatite by the Lu-Hf isotope system. In that way, I consider the manuscript a classic geological approach by using data from the micro/atom-scale to resolve the bigger picture of tectonics.

Relatively little is known about the Svecofennian orogeny (c. 2.0-1.8 Ga) in comparison to the other younger orogens dominating Europe, such as the Alps, Pyrenees, Variscan and Calededonides. To solve the complicated puzzle of this much older and metal-rich orogen, dating in combination with structural geology as presented in the concerned manuscript, is indeed very needed. Today, by far most of the published geochronological data from Fennoscandia comes from U/Pb on zircons, providing mainly the age of crystallization. Dating metamorphism and deformation within a minor error bar, however, is only possible recently and can be performed in only a handful of laboratories. In that perspective, the paper is useful for a large audience as it showcases a good example on how in situ dating of garnet can be applied to resolve for regional to crustal-scale tectonics.

In terms of locality, the authors have chosen their study area very right. Complicated but well mapped geology and directly relevant to society. The Olkiluoto study area has been highly investigated for the construction of a nuclear waste repository. The area represent the southwestern tip of Finland, which makes the correlation with the Swedish geology - mostly masked by the Baltic Sea - very appropriate.

The manuscript presents substantial conclusions which are in line with their own contribution and credited related work. The conclusions provide a solid base for future investigations, such as pinpointing study areas on the Swedish side and better constrain the evolution of the abundant crustal-scale shear zones.

As a layman on garnet dating, I still consider the applied scientific methods and assumptions clearly outlined. Only some technical terms may be explained a little more (see technical comments).

In combination with the provided supplementary data (shooting points and tables) the results are traceable for any fellow scientist.

When it comes to the title and abstract, I do miss the word "apatite". It is only in the introduction that I read for the first time that the authors applied Lu-Hf dating on both garnet *and* apatite. Is the latter less important? Please also elaborate throughout the manuscript on explaining the difference and the added value of dating both mineral types, in particular in relation to the cooling path/rate.

In general, the manuscript is well structured, well written and organized (only some technical comments).

***Response to Reviewer #1, Dr. Stefan Luth:***
Thank you for you review and comments to the manuscript. The main effort this study was focused on the garnet dating and apatite were only done as additional study. We have now added more information about this in the abstract and introduction, also how the coupling of the methods benefits the whole study. The title is also changed with addition of apatite in the title.

Comments in the manuscript by reviewer #1 in the manuscript:

Comment Line 112:

But typically there exists an interplay between both processes: Uplift creating erosion, and erosion creating uplift ;-)

This has been now rewritten.

Comment to Figure 2:

No fault? Just a lithological boundary?

No, not really a fault, and the contact is gradual from metaexite to diatexite.

Comment Line 134:

I try not to be too nasty, but what does this actually mean? How do you know? They do have gneissic banding and prominent stretching lineation.

This has been rewritten and added some information to be clearer.

Comment Line 245:

What is normal?

We have now named it a typical TGG rock in Olkiluoto and then the other type that is not typical, thus has an increased level of P. This is also now made clearer in the figure.

Comment Line 326:

?

This has been rewritten and added some information to be clearer.

Comment Line 383:

I am just wondering if there is any data published on the later cooling events in the area (below 400C). Such as with Ar/Ar. I might be useful to spent some lines on that?

Reference added regarding a recent paper on this, but generally the first dated brittle event after ductile-brittle transition is considerably younger so seems a bit out of the scope to elaborate more on that.

Comment to Figure 11:

This figure has been updated accordingly to suggestions. The grey shear zones are now only under the description of the various belts.

Comment Line 416:

I am a little confused here. Are both lines presented here as contrasting ideas? Or additional?

This text concerning Häme Belt and Olkiluoto site has been expanded and been evolved a bit more, however not totally clear which model, but the model, where a subduction followed by an extensional event caused by slab rollback event.

Comments Line 421:

the same as what? And Suggestion to refine sentence into "....of the garnets from this study are then interpreted to show the latter distinct.....

This text concerning the evolution and the interpretation of the age has been made clearer.

Comments Line 422:

Maybe you want to elaborate a little more on "the structural" if possible. Or/and refer to Engström et al. 2022.

Yes, this has now been expanded, with some more structural text from Engström 2022.

Comments Line 431:

You mentioned that this belt was affected by extension. can it be a core complex?
I have some thought/ideas that the Ljusdal batholite (1.87-1.84) might be.
I do not have any evidence suggesting that there is any batholite that have that appearance.

Comments Line 465:

I don't understand this part. I do after thinking a bit. Suggest to rewrite:
"why sedimentary units are mainly present offshore"
This has now been rewritten more clearly.

Comment Line 466:

Side note: this would be nice to investigate further in Sweden. There is a suspected N-S fault just offshore of Hudiksvall.

Yes, this would be very interesting!

Comment Line 476:

Also the the PT conditions appear similar for those parts of Finland and Sweden. Also note that in Sweden it is not only the time of M1 and M2 but also the time of crystallization of the large Ljudal Batholite, which comprises the bulk of the Ljusdal lithotectonic unit (see Högdahl and Bergman 2020)

The discussion concerning this has been expanded slightly more, and a collaborative study on both the Swedish and Finnish side would be very welcome.

Comment Line 495:

This one is a bit vague and not really an outcome as it is written know. Consider a rewrite.
This has now been rewritten a bit and some more information added about the study and coupling to the different belts.

Comment Line 499:

"crustal units" ?

Yes, good addition and clarification.

**Reviewer #2**

**Dear Editor and authors,**

**I appreciate the opportunity to review this manuscript, which demonstrates the application of the novel in-situ Lu-Hf dating technique of garnet and apatite in conjunction with U-Pb apatite geochronology to unravel the tectono-metamorphic evolution of the Olkiluoto site in southwestern Finland.**

**The authors mention only at the beginning of the manuscript that the findings of this study have the important goal to characterise the geological site which is planned to be used for the construction of a nuclear waste repository. However, it is not clear how such a site could be planned in a terrain of deformed migmatitic rocks and how the obtained findings can help characterising the site in light of this application.**

**Below are major comments that I believe will assist the authors in enhancing their manuscript. Additional comments, questions and notes are included in the attached PDFs (main text and supplementary figures).**

*Response to Reviewer #2:*
Thank you for you review and comments to the manuscript. The thorough review added several good points and enhanced the manuscript to make it clearer. The main effort this study was focused on the garnet dating and apatite were only done as additional study. We have now added more information about this in the abstract and introduction, also how the coupling of the methods benefits the whole study. The title is also changed with addition of apatite in the title. The study was made at the site for the planned nuclear waste repository and all information regarding site characterization is beneficial for the site. This study is just a little piece in the site investigations and was made possibly with the data acquired from the company in charge of the siting and will be the one implementing the procedures when nuclear waste will be buried at the site.

We have acknowledged all the comments to the manuscript and more detailed comments are found under each comment.

**Introduction:**

The introduction could be made more relevant and focused by addressing the integration of in-situ geochronology techniques for the key petrochronometers (garnet and apatite).

This part has been expanded and made clearer.

I recommend reshaping the introduction to expand on the new petrochronological tools employed to analyse the complex (potentially polymetamorphic) Precambrian terrains. This could involve incorporating part of the third paragraph that introduces the in-situ Lu-Hf dating technique for garnet, along with expanding on recent studies where this method was applied to ancient rocks, possibly together with other dating techniques, including Lu-Hf and U-Pb geochronology of apatite.

Yes, this has been expanded and the coupling between garnet and apatite Lu-Hf dating technique has been made clearer.

The discussion of apatite as a geochronometer is only introduced after line 49. Why is that? I believe that the integration of more petrochronometers is a powerful tool, and the authors should highlight this. Both in-situ Lu-Hf dating of garnet and apatite were recently developed (Simpson et al., 2021), and their integration for investigating complex metamorphic histories, along with other systematics, should be highlighted in the title, abstract, and detailed further in the Introduction.

We brought up this now clearer in the introduction. We have now added more information about this in the abstract and introduction, also how the coupling of the methods benefits the whole study. The title is also changed with addition of apatite in the title.

The tectono-metamorphic evolution of the Olkiluoto site could then be outlined in a subsequent paragraph. This can be streamlined by incorporating relevant information into the geological setting section.

Upon reviewing the manuscript, I noted the use of U-Pb dating of apatite and trace element analysis. These combined geochronological methods should be central to the Introduction, as they are crucial for unravelling the polymetamorphism of the studied site.

This has been added to the introduction.

**Geological setting of the study area:**

**2.1** I find this section partly repetitive, between the first two paragraphs. Those can be shortened and combined such that the reader can easily follow the already published geological overview of the region. I think that this section would benefit from combining the tectonic and the metamorphic events for the Svecofennian orogeny in one paragraph.

This has now been set into on paragraph to make it clearer.

**2.2** This section can benefit from being more concise and informative. Much of the information is vague and confusingly phrased.

This has been expanded, rewritten, and revised to be clearer.

**Methods of the study:**

This section should be titled "Methodology" or "Petrochronology Approach", as it includes the sample descriptions. Alternatively, the investigated samples could be described before Section 3 which can then focus on the analytical methods employed.

This is now changed to Methodology and all aspects from outcrop to sampling and analytical methods added with different subsections.

**Sample description:** It would benefit from including a description of the relationships between the compositional layering and the leucosomes, helping the reader connect hand-sample information to the micro-textures. This is crucial because garnet crystals within the matrix and leucosomes were dated, and most interpretations rely on this data. The authors should refer to Fig. 4 in the text.

This has been expanded and made clearer connections to Fig. 4 in the text.

**EPMA:** The minerals analyzed for this study should be specified in this section. A table presenting representative mineral chemistry analysis should be included in the main body of the manuscript for better access to this information, which is relevant to this petrochemical and petrochronological study.

Since the paper has lot of different methods and this part of the study is not main part of this paper, we think that this table can be as part of the supplementary material since it is online and only a click away when you read the paper. However, if the Editor seem this is crucial to have in the paper we can add it.

**In-situ Lu-Hf geochronology of garnet and apatite:** Methodologies for calculating uncertainties should be incorporated in the main text, and all dates obtained for the reference materials must be reported.

This has been added.

**Method – Bulk-rock geochemistry:** This subsection is missing from the methods section.

This has been added.

**Texture-geochronology-REE-PT modelling:** U-Pb geochronology and trace element data of apatite were collected along with in-situ Lu-Hf dating of garnet and apatite. However, this method is not mentioned in the

text until line 207. The authors should clarify why both Lu-Hf and U-Pb geochronology were employed to date apatite. This should also be reflected in the abstract and Introduction. More textural information about the analyzed apatite grains should be provided in the manuscript, accompanied by BSE images of the grains (at least in the supplementary materials), the number of grains analyzed, and the sample IDs from which the apatite grains originated. Apatite was described only as inclusions in the rim of a large garnet crystal (sample MM31), whereas the P-T modeling was conducted on sample MM30A, which does not describe any apatite crystals nor report them in this manuscript (Fig. 5B). I want to emphasize that the scattered analysis, observations, and descriptions between the two different samples (MM30A and MM31) are continuously mixed throughout the manuscript, both in the results and in the discussion sections, leading to interpretations that lack robust data support. Meso-scale and microscale textural observations of garnet textures and associated mineralogy appear different between the two samples; therefore, I find it hard to use the information from the two samples in an interchangeable way. Finally, whole rock geochemistry is presented for only one sample, which limits the geochemical comparison between the two.

The text has been expanded and made clearer on this point. The sample description concerning the two samples has been more consistently described throughout the paper.

Small garnet crystals are reported within the matrix, defining compositional layering in the studied TGG (sample MM30A), while a larger garnet crystal is enclosed within felsic leucosomes in sample MM31. It is crucial for the authors to better describe and report the relationships between the leucosomes and the dominant foliation, considering the interpretations are based on garnet data obtained from these two features. These two types of garnet crystals also exhibit different isotopic and REE analyses, indicating at least different ages, Lu content, and textural contexts. Therefore, while the uncertainties for Lu-Hf dates of garnet crystals do not allow for confident interpretations, data should not be consolidated into a single, individual age derived from different garnet crystals and samples, especially given their distinct textural contexts.

The distinction between the small garnets and big garnets have been described more clearer and thus now it should be easier to follow the procedure concerning dating of those and how apatite has been used during this study. The uncertainty regarding the garnet ages has now been described more and the age data for the big garnet and the small garnet are presented. However, we argue that the Lu-Hf dates can be consolidated to a single age data due to several reasons that is now explained in the paper. A few of those are the largely flat major element zonation of both large and small garnets with similar absolute values. This would indicate that both small and large garnets equilibrated at the same high T conditions. Thus, we interpret that the large and small garnets grew in response to one metamorphic event, that would be the latter metamorphic peak in the Olkiluoto site.

**Discussion:** Most of my comments on this section are included in the attached PDF. I find this section poorly structured; the textural and petrochronological observations, along with P-T modeling of the two samples, are merged and used for speculative interpretations unsupported by the data. Many paragraphs contain lists of metamorphic ages recorded elsewhere within the extensive 40,000 square kilometre region shown in Fig. 11, and where their relationships with the studied site are unclear.

The discussion has now been rewritten especially in the beginning to address more clearly the aim of the study, the results in clearer context and the connection to the especially the Häme belt. The authors are aware of the huge 40,000 square kilometre region shown in Fig. 11 and acknowledge that the figure only shows some summary of ages for different belts. However, this also highlights the importance of this kind of studies where some structural geological mapping and new novel metamorphic mineral age data (Lu-Hf Garnet and Apatite age techniques) needs to be more use in the future especially for this old crustal units with polymetamorphic environment.

**Overall**, I encourage the authors to be more specific and less vague throughout the text to improve clarity and readability. Samples are often referred to without their sample IDs, merely identified as "rock." I also advise employing a labelling system for the analyzed garnet crystals, such as "garnet 1" (the large crystal in sample MM31) and "garnet 2," "garnet 3," and "garnet 4" for the smaller garnet crystals in sample MM30. This labelling scheme will help the reader easily track which analyses correspond to which sample and grain.

This labelling has now been implemented throughout the paper. This should make it easier for the reader to understand how the small and big garnets have been analysed and how the interpretation has been done.

**Supplementary material:** The supplementary material should be labelled as Table S1, Table S2, Table S3, etc., to clearly indicate to which Supplementary Table the authors are referring. I recommend removing the "Index" label in the Supplementary Data Table and instead labelling each table as S1, S2, etc. Each table should have a caption that describes its content.

I invite the authors to carefully revise the manuscript, particularly regarding interpretations based on merged dates obtained from different samples and petrochronometers with varying textural contexts.

The supplementary material has also been made more clearer and the reference to the various supplementary figures and tables are made easier to follow.

***Main Comments in the manuscript by reviewer #2 in the manuscript (excluded those concerning small text adjustments and already addressed above):***

Comment Line 86:

are these events different from the 1.84-1.80?

This section has been modified and shortened as suggested.

Comment Line 94, 97, 98 and 99:

which one are those? are these part of the supracrustal rocks? which rock-type? Add reference to age?

This section has been modified and relevant text added, the reference to age of the dykes has also been added.

Comment Lines 105-107:

what about the temperature? It is not clear to me what is meant here? What are the PT conditions of the first metamorphic event and what is its geodynamic significance?

This section has been modified and relevant text added to make this a bit clearer, unfortunately not all information available therefore the previous research is a bit vague.

Comment Line 110:

but what about the temperature for the first metamorphic event?

This information is not available and the difference of 2 kbar clarified more in text.

Comment Figure 2:

What about all this structural data? They are never mentioned in the text. They are not relevant to this study. A better characterization of the studied outcrop would be more informative to the reader.

The outcrop has been described more thoroughly text and more information can be found in Engström et al. 2022. The geological map has been made clearer since a simpler structural is better as suggested by the reviewer. The outcrop is well imaged in Figure 3, thus we think it is a sufficient description and image for this study more focused as a metamorphic study.

Comment on Method section:
Methods include only analytical methods. If the sample description is included than this should be modified to Methodology or the Sample/outcrop description should be a section before the Analytical methods.

This has been modified.

Comment Line 120-124:

Cannot be the protolith of a metamorphic event but of a metamorphic rock, please rephrase?

This whole section has been modified and thus made clearer for the reader.

Comment section 3.1 (now 3.2) Sample description Line 134-154:

this is not a description but an interpretation. Here the linear and planar fabrics should be described along with mineralogy and field/hand-sample relationships?

here the relationship between the matrix and the lecosome where the different types of garnet were investigated should be described? Refer to Fig 4 and Fig 5

This section has now been modified accordingly following these comments from reviewer. It should now be easier to follow in the text the different samples and what have been done with them. Also the coupling to figures added.

Comment Line 166:

I see that the supp material includes analyses on many other mineral phases. All the phases analysed by EPMA should be reported in the text and the supp must be referenced!

The connection to supplementary material has now been made more evident and reference to exact supplementary tables added.

Comment section 3.3 (now 3.4) Garnet and apatite in situ Lu-Hf geochronology Line 183:

Sorry, but the way the Supp Data Table is presented is unacceptable. Tide it, every Tab/Table must have a caption, "x" (I guess omission) should have notes/comments that explain their omission etc etc.

We agree this has been made tidier. Within the text we have also added direct reference to the supplementary tables and exactly which table it concerns.

Comment Line 184:

What about apatite? How were the samples for apatite prepared? Were the apatite grains analysed the one included in garnet crystals? If so please specify this

This has been included.

Comment Line 198, 214 and 219:

specify here the age produced by the analysis of Högsbo? specify here the primary RM used for garnet isotope and trace element analysis?
for which mineral?
Here the authors must specify how uncertainties were calculated?
Yes, all these comments have been acknowledged and relevant information has been added.

Comment Line 230:

Ti2O is contained in biotite which is relatively abundant in the sample. Did the authors try to include it in the calculation to check whether any significant PT variations are shown by its incorporation?

PT modelling using mineral assemblages is more reliable than mineral compositions so the presence of Ti bearing phases such as ilmenite and rutile would have necessitated the addition of $TiO_2$ to the system. Ti content in biotite was not investigated in our modelling - we have chosen to use garnet since it is the dated mineral, giving us PT constraints and ages. The work of Douce 1993 indicate that Ti does not affect the stability of biotite.

Comment Line 242:

which metamorphic constraints?

This has been rephrased.

Comment Line 244:

The diabase dykes form the Olkiluoto region were excluded from the dataset, due to?

Why these were excluded have now been stated.

Comment Line 245:

Concerning TGG rocks, normal? what does it mean?

These are divided into two different types the typical TGG and the ones with elevated amount of P. The difference is explained in the next line and how much they differ from each other.

Comment Figure 6:

this figure is very hard to read. Use different color for the two symbols in both graphs!

Yes, this has been modified.

Comment Heading 4.2:

major and trace element mineral chemistry?

This has been modified.

Comment Line 270:

this sentence contraddicts the previous where garnet rim of the big crystal (garnet 1) should contain less Y which is instead incorporated by apatite growth in garnet rims. This is supported by the Supplementray Table (Garnet Lu-Hf data) showing higher Lu content in the core relative to the rim of the big crystal?

This has been explained and added to the text.

Comment Line 272:

what about the foliation? It seems that biotite is aligned parallel to the compositional layering. Why omitting this information?

This has been added.

Comment Line 275:

How is this information useful? Likely a textural description of which texture Kfs exhibits is more useful to the reader.

This just a description of the samples and stating that they are similar concerning K-feldspar.

Comment Line 293:
out of 84 analyses 2 were excluded, yet in the Fig. 8A 83 analyses are reported?

Yes, this has been fixed and explained in the text. There is one duplicate sample.

Comment Line 296:
are these uncertainties 2 SD absolute?

Yes, they are, and some uncertainties has accordingly been adjusted.

Comment Line 305:
texturally, would this make sense?

This is explained now in more detail in the text. Structurally these ages could be interpreted as two different types of garnets. However, the garnet Lu-Hf age data does not distinguish them as two generations. The method and uncertainties overlap each other so we decided to keep the data plotted together. Below you find the exact formulation from the manuscript:

*The small garnet grains were targeted with 51 analyses in total (see Supplementary Fig. S6 for spot locations) and have a restricted $^{176}Lu/^{176}Hf$ ratio range, resulting in an anchored isochron age of 1857 ± 49 Ma (n=51; MSWD = 1.1; Fig. 8B). If the grains are plotted separately, they all produce the same age within uncertainty, but due to the smaller number of analyses the errors are larger. The small garnets have Lu contents ranging from 10 ppm to below the detection limit (average is 4 ppm). Given the overlap in uncertainty, the proximity and similarity of the samples, if all the garnet data is plotted together and anchored to an initial $^{177}Hf/^{176}Hf$ ratio of 3.55 ± 0.06, an isochron age of 1834 ± 17 Ma (n=134; MSWD = 1.2; Fig. 8C) can be produced.*

Comment Line 307-308:
No information about how many apatite grains were dated, neither their textural context is provided. Is there apatite in the matrix? Are the apatite grains limited to garnet rims of one garnet crystal in MM31? Why? the authors never explain why they used two systematics to date the same mineral.

This section has been expanded and more details regarding Lu-Hf and U-Pb geochronology on apatite has been added.

Comment Line 326:

based on what? not clear?

This was not clearly written. This has been re-written.

Comment Figure 10:

Use kbar! which one is what? Use different colors and/or dashed and continuous lines to show the different compositional isopleths? use different colors for each compositional isopleths. As now, it is not understandable which area is what?

The figure has been refined accordingly to the comments from reviewer.

Comment to discussion and part 5.1 Significance of the age data and metamorphic constraints:

More should be expanded on the textural, chemical and isotopic results that lead the authors to combine all Lu-Hf data of garnet in one single age (considering that the data is not only from different grains with different textures, but also different samples. is that supported by meso and micro observations? IDates from two different samples cannot be combined in one single age, considering also that the garnet crystals have a different textural context, different chemistry and grew within different structural and litholgical context.

Our interpretations and discussions have now expanded considerably concerning this issue. Below you find the exact formulation from the manuscript:

*These results are statistically indistinguishable, meaning that if there is a difference in age between the small and large garnets, then it is not possible to define this with the Lu-Hf method. When all the Lu-Hf data are plotted together, it is clear that the large garnet data controls the position of the isochron resulting in a combined age which is identical to*

*the large garnet age (1834 ± 17 Ma and 1829 ± 19 Ma respectively). Texturally, the small garnets occur in the rock matrix, while the larger garnets occur in the leucosome. Plausibly, this may mean that the smaller garnets are older. However, both the sample matrix and leucosomes are deformed by the same foliation (Figs. 3, 4, 7) suggesting it is equally feasible that all garnet grew in response to a single phase of deformation.*

Comment Line 355:

such as ? indicate c. temperatures in °C? how do the authors then justify the different Lu content between core-rim and small grains?

The temperature has been added. The leucosome garnets growing in melt may have had a larger equilibrium volume due to more effective transport of elements within the melt resulting in the highest Lu contents in the grain cores (average of 38 ppm) with lower values in the rim (average of 21 ppm) following garnet (and melt) crystallization.

Comment Line 356:

high-grade? The PT modelling yields temperatures of c. 700 °C, which are lower than the closure temperature for the Lu-Hf system of garnet (700-900°C)

This has been re-written from high-grade to upper amphibolite facies metamorphism. This part has also been expanded so that we justify the interpretation.

Comment Line 363:

but are they? I think that the symplectitic qurtz played probably a bigger role than the authors are reporting. The apatite crystals enclosed within the garnet rim (and a couple in garnet cores) do not seem to be in contact with the matrix, rather they seem to be always associated with the qtz symplectite.

Other garnets clearly have apatite that is in contact with the matrix - we only have a 2D image of this grain, so it is totally possible they are in contact with the matrix (at least one of the dated grains definitely was in contact with the matrix because it occurs on the edge of the grain). We have no problem indicating that symplectitic quartz may also have played a role though.

Comment Line 365:

if the peak is around 700°C how can the sample then stay at higher temperatures for 50 Ma? textural /mineral evidence would show that.

This would be consistent with the relatively flat major element compositional profiles in garnet. Producing nearly flat profiles in a large garnet grain would require a prolonged period (likely on the order of 10s of millions years) at temperatures above 650 °C.

Comment Line 368 and 370:

specify that these pegmatites formed during the migmatization (partial melting) event! pegmatitic intrusion? or migmatization? These are ages consistent with the dates obtained in this study and interpreted as ages for the high-temperature partial melting event.

This has now been clearly written, the context was vague before.

Comment Line 372:

of what? which rocks? and how are those related to the studied samples/site?

This has now been re-written and defining that the previous study was performed in the Olkiluoto site.

Comment Line 374:

As previously commented the compositional isopleths cannot be distinguished. Please modify them in order to be clear for the reader

The figure has been modified accordingly.

Comment Line 378, 380 and 384:

Although the two obtained garnet ages are within uncertainty this does not mean that they can be pulled together. The small garnet crystal indeed are texturally within the matrix and yield an older age than the big garnet crystal that grew in the leucosome. Apatite yield even younger ages, likely related to the final stages of the partial melting event in association with late hydrothermal fluids. I disagree in combining garnet dates obtained from two different samples as one age and compare it with apatite ages from only one sample.

this is completely speculative. Are there any study in the region about the exhumation stage?

We have defined previously the reasoning why we prefer to combine these ages from garnet Lu-Hf age data together. We have now re-written the tectonic evolution so the garnet and apatite Lu-Hf age data would have a plausible explanation.

Comment Line 388:

I would like to see FIRST a summary figure where textural and mineralogical observations are combined with the textural context of the used petrochronometers and the PT conditions for the investigated sample(S).

We feel that this would not be added value to the paper since we now explained better our reasoning in the text. Also, since limited amount of this kind data is at available at the moment in the region a comparison would therefore be hard to perform.

Comment Line 389:

the two mentioned metamorphic facies include very different PT field. Is this a PTt loop for the Tampere Belt? If so report T and P such that the reader can compared with the data reported in this study.

This has been clarified better in the text now. No P-T loop reported for Tampere belt. The different metamorphic belts have general previous studies done with a different approach thus a more comprehensive study with new techniques would be interesting and needed.

Comment Line 391:

What is meant with high grade conditions?? Please be more specific ion the assemblage garnet was stable with and the PT conditions reported in that study. Is this study also referred to the Tampere Belt? More sense between the sentences could be made if the authors make clearer what are we comparing? what type of rocks'? we cannot just compare random ages and PT conditions from random rocks in such a huge region! This style of writing is very frustrating.

We acknowledge that the structure of this section was not structured well. The section has now been re-written considerably and thus giving more insight to the complex structural and metamorphic environment of southern Finland.

Comment Line 395:

what is it, a metamorphic gradient? some domain of this belt are low grade upper crustal rocks and others are mid-to lower crustal rocks? peak conditions of what?

This has been made clearer in the text and defining that different areas have different metamorphic signature.

Comment Line 396:

I find this part very uninformative. It sounds like a list of metamorphic facies and random ages. How are these supposed to be useful to better constrain the tectono-metamorphic history of the Olkiluoto site? At least describe which rocks (including mineral assemblage) were dated (which mineral was dated), how they can relate to the investigated samples and the obtained data. Although, no structural work was performed in this study, previous work should be used to integrate the metamorphic information with the crustal-scale structures of this part of SW Finland.

We acknowledge that the structure of this section was structured poorly. The section has now been re-written considerably and thus giving more context to tectonically and metamorphic constrains and coupling of the Olkiluoto site with the Häme Belt further east from the site.

Comment Line 415 and 417:

in the first the metamorphic facies is given, wheres for the second "high-T event ". Can the authors be more informative on the OPT conditions for these polymetamorphic history? I don't understand. The first authors are trying to interpret the two distinct high-T event as a prolonged anatexis event (not supported by this data). It is not cleat how Torvela amd Kurhila 2022 explain both events with one late event only.

I don't understand. The first authors are trying to interpret the two distinct high-T event as a prolonged anatexis event (not supported by this data). It is not cleat how Torvela amd Kurhila 2022 explain both events with one late event only.

The section has been re-written to be clearer.

Comment Line 420:

no structural studies have ever been mentioned and incorporated in the interpretation yet in the text of this manuscript.

Some text concerning structural studies has been added in the text, but still, we prefer to refer to Engström et al., 2022.

Comment Line 429:

Not clear how roundish qz-fsp megacrysts are related to a prolonged cooling of diatexitic migmatites. Please elaborate.

This interpretation has now been expanded and some references how the connection might be deduced added.

Comment Line 430:

The authors mentioned that this belt underwent extension. these deeper crustal levels may have been exhumed along low angle detachment zone. discuss this to better link tectonic regimes and metamorphic events in the region.

This is referring to Uusimaa Belt that possibly underwent extension. However, we consider this a bit distal to our study area so we would think this is out of the scope for this study.

Comment Line 430:

can the authors refer to T/P ratio? (see Brown and Johnson collection of papers on metamorphism through geological time)

Geothermal gradients for different areas have been added to the text.

Comment Line 453:

this is not a structural study and no structural information support any of the data in this contribution.

It is true that this is not a structural study but in discussion we consider it is feasible to highlight that also this aspect is important.

Comment Line 495:

This study does not provide any additional structural constrain to the existing one.

This conclusion has been modified.

---

## Referee Report (RR1)

**Insights into the tectonic evolution of the Svecofennian orogeny based on *in situ* Lu-Hf dating of garnet and apatite from Olkiluoto, SW Finland**

by

JON ENGSTRÖM, KATHRYN CUTTS, STIJN GLORIE, ESA HEILIMO, ESTER M. JOLIS AND RADOSLAW M. MICHALLIK

**General**

This manuscript has been previously reviewed by two experts and I have had an opportunity to see their assessments. Therefore, I have tried not to repeat their comments. However, because I agree with the notion of Reviewer 2# that the samples and their analytical results should be treated separately, I added my views to the same topic. While reading, I also made some small formal comments on language etc., they can be skipped if inappropriate.

The main topic here is dating of different phases of metamorphism by in-situ Lu-Hf method on garnet and apatite. The latter was also dated by U-Pb method. This method is quite novel and therefore, as far as I understand, still in initial stage. Therefore, the results may contain shortcomings that are not yet fully understood. By this I mean e.g., the very large errors the method yields. Or maybe the instruments used are not yet sensitive enough for high resolution.

The garnet dating from leucosome and mesosome failed to decipher the two metamorphic events that were previously dated by U-Pb method from zircons in leucosomes. All the garnet analyses combined gave a "pooled" age of $1834 \pm 17$ Ma. Meaningfulness of such approach can be questioned. This is further discussed below in detailed comments. Despite this quite critical review, I see this method promising and at this stage even the difficult cases are good to publish for the use of future developments.

The apatite was dated by in-situ Lu-Hf and U-Pb. These data are highly welcome to regional metamorphic studies as they are rarely used in this part of the Precambrian world.

**Detailed comments**

line 26-30: repetitive use of the word "polymetamorphism"

31-34: There is a bold statement arguing that Lu-Hf garnet method represents the metamorphic age of the rock better than the U-Pb monazite or zircon. This can be questioned. Lu-Hf garnet is still quite a new and incompletely understood method which so far gives quite large errors. U-Pb zircon in leucosome is a widely used and well-known method to date peak metamorphism (it is assumed that melting is coeval with peak metamorphism and zircons crystallise in the melt, hence the connection) with high precision. Monazite is another story…

38-39: It is stated that "at least one, possibly two, significant metamorphic events". According to previous investigations, it is quite convincingly shown that there are at least two metamorphic events in the region. I will come back to this this later while commenting Chapter 5.1.

83-84: Yes, crustal thickening took place, but neither of the cited articles argued that it caused the peak metamorphism and melting. Instead, Mäkitie et al. stated that the heat source is unknown and Chopin et al. said that the melting is related to channel flow.

90-92: Transpressional: Ehlers et al. 1993 (Precambrian Res.) were the first to describe the transpressional tectonics in this part of the world; should be cited.

102. No good to start a chapter with a reference. Change the word order.

123-125. About the two metamorphic events, see comment line 390.

137: "This study includes whole rock geochemistry of the different lithological units". See my comment on line 272.

138-139: "in the first metamorphic phase in Olkiluoto (Engström et al., 2022)". Because these two phases are repeatedly mentioned, it could be appropriate to call them, e.g., the first metamorphism (M1, older) and the second (M2, younger).

145-146: In this ms. a migmatitic rock is divided into the leucosome and matrix. In the common migmatite terminology these are called the leucosome and mesosome.

158-160: In this sample description it clear that different types of garnets occur in the mesosome vs. leucosome. See later comment.

180: Chapter 3.3. These analytical methods are already published in the cited report which is openly available online.

188: Do you mean "The second largest" ?? or maybe "Another large garnet" ??

272: Chapter 4.1. The whole chapter is a little bit strange for several reasons. The authors do not have own data but refer to the data in Kärki & Paulamäki (2006) report, which is already published and available online. The data tables are not in the report, so this maybe is what the authors mean in the Acknowledgements by expressing their gratitude to Posiva Oy for access to the data. Here the authors only use some major elements and show the TAS and AFM diagrams. The TAS diagram was also shown in the original report (but not the AFM diagram). These data are not used in the later discussion. The data seem to have no value for the topic of this ms., at least it is not described.

276: …then…?

324: Chapter 4.3. These garnet ages are the core of the ms. I will come back to these later in Discussion.

328: …the age calculations…

328-229: …When all the data are plotted…

332 and 334: Give the MSWDs also for the core and rim ages. I explain later why on line 390.

334: not identical age, just overlapping within errors.

341: "Given the overlap in uncertainty, the proximity and similarity of the samples," This sentence can be removed, these are already described earlier and complicate the sentence, when it continues with "if"

342: …data are…

347: (…MSWD = 1.5). According to Fig. 9A it is 1.6

348-349: These ages are not identical, just overlapping within errors.

353: …isochron age -> lower intercept age

390: Chapter 5.1. In this chapter the age determinations are discussed. In the earlier paper from the same locality (Engström et al. 2022), the metamorphism was dated by U-Pb zircons from two leucosomes in the migmatitic metapelites as $1858 \pm 7$ Ma and $1851 \pm 8$ Ma. As no younger zircons were found, the leucosome must have been related to the first melting episode, M1 (cf. Saukko et al. 2020). The second metamorphism M2 in Engström et al. (2022) was less precisely defined at 1.82-1.78 Ga. Just a few kilometres S of Olkiluoto in Rauma, Vehkamäki et al. (2021, Inst. Seism., Univ.Helsinki, Rep. S-71) dated zircons of two populations from the leucosome in a migmatitic metapelite; 1.86 and 1.83 Ga. Towards the W in central Sweden Högdahl et al. (2012) also dated two metamorphic events at 1.87-1.86 Ga and 1.82-1.80 Ga. The younger M2 is ubiquitous in southern Finland and central Sweden, described in numerous studies. It is the older M1 that has only lately emerged from behind the strong M2 overprint.

On the basis of the previous age determinations reviewed above, it is very likely that these events also prevailed in Olkiluoto, as proposed by Engström et al. (2022). In the present study, the authors have dated two types of garnets; small garnets in the mesosome and large garnets in the leucosome. In the sample description and also later it is speculated whether the garnets are of different generations. In fact, the face value age (without error) for the small garnets is 1857 Ma and 1829 Ma for the big garnet, which is further divided into 1828 ($\pm11$) Ma for the core and 1829 ($\pm21$) Ma for the rim domains. It is a pity that the MSWDs for these analyses were not shown, but I guess is that the core age has the lower one. Looking at these ages without errors, they are exactly what might be expected for the M1 and M2. Now the problem is the very large errors in the ages. To solve this the authors chose to use a pooled age of all the analyses to get smaller errors (result 1834 $\pm$ 17 Ma). This is a highly questionable way to use the data. It seems that the sampling was not structurally controlled to separate possible different garnet generations. To the S of Olkiluoto in the Turku area, two generation of garnets were described, the first elongated syn-D2 garnet was deformed in D3 which in turn was coeval with leucosome containing large garnets (Väisänen & Hölttä 1999; Fig. 14g). This might be the case in Olkiluoto, too. But the major problem here is not the sampling, but the resolution of the Lu-Hf garnet method, which is incapable of solving so detailed a problem.

392: …all the analyses…

394: …, …

396: these ages are not identical, just overlapping within error

409: …the(se) data form…

417: …in the U-Pb and…

426: …The zircon U-Pb…

475: Fig. 11. Why is the Turku granulite area skipped from the examples in Figure. It is anyway isotopically and metamorphically well-studied migmatite area close to Olkiluoto.

499: …in the Häme Belt…

509: "reaching slightly deeper crustal depth than the Häme Belt" ? what does this mean?

512: …the Uusimaa Belt…

513: …the anatectic melts (or melting)

514-515: "indicates that the whole southern Finland domain was subjected to a long hot orogenic evolution with several crustal-scale melt pulses". Chardon et al. 2009. Tectonophysics 477, describes this nicely, worth citing

517: Chapter 5.3. This chapter shortly summarises the present knowledge of the chapter title, focusing on the shear zones, but it hard to understand how this literature review is connected to the data in the ms., garnet and apatite age determinations.

519 and 521: back-arc vs backarc … Be consistent with writing style

528: Additionally, additionally what? unclear

531: …show that…

574: "similarities between the Olkiluoto site and Häme Belt, and the differences to Uusimaa Belt." What actually are the real differences between the Häme and Uusimaa belts? So big that they are part of the main conclusions? This topic is not clearly discussed in the ms. There are evidently differences in erosion levels, but what else? If this refers to the metamorphic ages, two metamorphic events are found also in the Uusimaa Belt, 1.86 and 1.83 Ga (Väisänen et al. 2021, Inst.Seism., Univ Helsinki, Rep. S-71). The protolith ages are also the same, 1.86 Ga.

623: References: Many references occur twice in the list, e.g., Lahtinen et al. 2005, and many others. Delete the extra.

624: page numbers missing

714: page numbers missing. Check all the similar references

796: Pitkälä…This is an unpublished MSc thesis. A published document should be used: Pitkälä et al. 2018. Inst.Seism., Univ Helsinki, Rep. S-67.

6.11.2024

---

## Author Response (AR2)

**REVIEW RESPONSES TO MANUSCRIPT VERSION 2**

Please find our revised manuscript titled "***Insights into the tectonic evolution of the Svecofennian orogeny based on in situ Lu-Hf dating of garnet and apatite from Olkiluoto, SW Finland***" that we submit for consideration to be published in the Solid Earth.

Below you also find comments to the review.

Sincerely on behalf of all authors,

Jon Engström

Reviewer #3

**Insights into the tectonic evolution of the Svecofennian orogeny based on *in situ* Lu-Hf dating of garnet and apatite from Olkiluoto, SW Finland**
by
JON ENGSTRÖM, KATHRYN CUTTS, STIJN GLORIE, ESA HEILIMO, ESTER M. JOLIS AND RADOSLAW M. MICHALLIK

*General*
*This manuscript has been previously reviewed by two experts and I have had an opportunity to see their assessments. Therefore, I have tried not to repeat their comments. However, because I agree with the notion of Reviewer 2# that the samples and their analytical results should be treated separately, I added my views to the same topic. While reading, I also made some small formal comments on language etc., they can be skipped if inappropriate.*
*The main topic here is dating of different phases of metamorphism by in-situ Lu-Hf method on garnet and apatite. The latter was also dated by U-Pb method. This method is quite novel and therefore, as far as I understand, still in initial stage. Therefore, the results may contain shortcomings that are not yet fully understood. By this I mean e.g., the very large errors the method yields. Or maybe the instruments used are not yet sensitive enough for high resolution.*
*The garnet dating from leucosome and mesosome failed to decipher the two metamorphic events that were previously dated by U-Pb method from zircons in leucosomes. All the garnet analyses combined gave a "pooled" age of 1834 ± 17 Ma. Meaningfulness of such approach can be questioned. This is further discussed below in detailed comments. Despite this quite critical review, I see this method promising and at this stage even the difficult cases are good to publish for the use of future developments.*
*The apatite was dated by in-situ Lu-Hf and U-Pb. These data are highly welcome to regional metamorphic studies as they are rarely used in this part of the Precambrian world.*

**Dear reviewer,**

**We appreciate all additional comments to the manuscript that improved this second version of the manuscript. We acknowledge especially the comments on the Lu-Hf method and thus we have been additionally modified the text according to the comments. Since the small garnets (MM30), have low Lu content for the garnets that we sampled, we decided to not "pool" the age and thus in the final discussion of the results we use mainly the large garnets (MM31) Lu-Hf ages.**

The response to the detailed comments is answered and found below with red text next to the comment.

**Detailed comments**

line 26-30: repetitive use of the word "polymetamorphism"
This has been accordingly adjusted.

31-34: There is a bold statement arguing that Lu-Hf garnet method represents the metamorphic age of the rock better than the U-Pb monazite or zircon. This can be questioned. Lu-Hf garnet is still quite a new and incompletely understood method which so far gives quite large errors. U-Pb zircon in leucosome is a widely used and well-known method to date peak metamorphism (it is assumed that melting is coeval with peak metamorphism and zircons crystallise in the melt, hence the connection) with high precision. Monazite is another story…
This statement has been re-written and modified that is a new method that is a good complement to older methods of age determination, such as U-Pb zircon dating.

38-39: It is stated that "at least one, possibly two, significant metamorphic events". According to previous investigations, it is quite convincingly shown that there are at least two metamorphic events in the region. I will come back to this this later while commenting Chapter 5.1.
This has been modified and defined now from the previous studies that two events occurred.

83-84: Yes, crustal thickening took place, but neither of the cited articles argued that it caused the peak metamorphism and melting. Instead, Mäkitie et al. stated that the heat source is unknown and Chopin et al. said that the melting is related to channel flow.
This has been accordingly modified.

90-92: Transpressional: Ehlers et al. 1993 (Precambrian Res.) were the first to describe the transpressional tectonics in this part of the world; should be cited.
This citation has been added.

102. No good to start a chapter with a reference. Change the word order.
This has been accordingly modified.

123-125. About the two metamorphic events, see comment line 390.
This has been accordingly modified.

137: "This study includes whole rock geochemistry of the different lithological units". See my comment on line 272.
The motivation to add this geochemistry part to the manuscript has been added.

138-139: "in the first metamorphic phase in Olkiluoto (Engström et al., 2022)". Because these two phases are repeatedly mentioned, it could be appropriate to call them, e.g., the first metamorphism (M1, older) and the second (M2, younger).
This has been accordingly modified.

145-146: In this ms. a migmatitic rock is divided into the leucosome and matrix. In the common migmatite terminology these are called the leucosome and mesosome.
This is a valid terminology change and has now been implemented in the whole document.

158-160: In this sample description it clear that different types of garnets occur in the mesosome vs. leucosome. See later comment.

Yes, that was the aim of the sampling and hopefully now better described.

180: Chapter 3.3. These analytical methods are already published in the cited report which is openly available online.
Yes, but an earlier revision asked for this, so we have short recap about that.

188: Do you mean "The second largest" ?? or maybe "Another large garnet" ??
This has been changed accordingly, it is another large garnet from the same leucosome that has been used.

272: Chapter 4.1. The whole chapter is a little bit strange for several reasons. The authors do not have own data but refer to the data in Kärki & Paulamäki (2006) report, which is already published and available online. The data tables are not in the report, so this maybe is what the authors mean in the Acknowledgements by expressing their gratitude to Posiva Oy for access to the data. Here the authors only use some major elements and show the TAS and AFM diagrams. The TAS diagram was also shown in the original report (but not the AFM diagram). These data are not used in the later discussion. The data seem to have no value for the topic of this ms., at least it is not described.
This reason to include this part in the manuscript is to describe the chemistry of the TGG rock type in Olkiluoto and thus have a more regional coupling to the rocks outside Olkiluoto. This reasoning is also coupled now more to section 5.2.

276: …then…?
This has been accordingly modified.

324: Chapter 4.3. These garnet ages are the core of the ms. I will come back to these later in Discussion.
This is explained later in the discussion.

328: …the age calculations…
This has been accordingly modified.

328-229: …When all the data are plotted…
This has been accordingly modified.

332 and 334: Give the MSWDs also for the core and rim ages. I explain later why on line 390.
These were included in the supplementary figures referenced in the text but have now also been added here.

334: not identical age, just overlapping within errors.
The core age is 1828 ± 11 Ma and the rim age is 1828 ± 21 Ma – this is an identical isochron age but with a larger uncertainty so no change has been made here.

341: "Given the overlap in uncertainty, the proximity and similarity of the samples," This sentence can be removed, these are already described earlier and complicate the sentence, when it continues with "if"
This has been accordingly modified.

342: …data are…
This has been accordingly modified.

347: (…MSWD = 1.5). According to Fig. 9A it is 1.6
This has been corrected.

348-349: These ages are not identical, just overlapping within errors.

This has been accordingly modified.

353: …isochron age -> lower intercept age
This has been accordingly modified.

390: Chapter 5.1. In this chapter the age determinations are discussed. In the earlier paper from the same locality (Engström et al. 2022), the metamorphism was dated by U-Pb zircons from two leucosomes in the migmatitic metapelites as 1858 ± 7 Ma and 1851 ± 8 Ma. As no younger zircons were found, the leucosome must have been related to the first melting episode, M1 (cf. Saukko et al. 2020). The second metamorphism M2 in Engström et al. (2022) was less precisely defined at 1.82-1.78 Ga. Just a few kilometres S of Olkiluoto in Rauma, Vehkamäki et al. (2021, Inst. Seism., Univ.Helsinki, Rep. S-71) dated zircons of two populations from the leucosome in a migmatitic metapelite; 1.86 and 1.83 Ga. Towards the W in central Sweden Högdahl et al. (2012) also dated two metamorphic events at 1.87-1.86 Ga and 1.82-1.80 Ga. The younger M2 is ubiquitous in southern Finland and central Sweden, described in numerous studies. It is the older M1 that has only lately emerged from behind the strong M2 overprint.
On the basis of the previous age determinations reviewed above, it is very likely that these events also prevailed in Olkiluoto, as proposed by Engström et al. (2022). In the present study, the authors have dated two types of garnets; small garnets in the mesosome and large garnets in the leucosome. In the sample description and also later it is speculated whether the garnets are of different generations. In fact, the face value age (without error) for the small garnets is 1857 Ma and 1829 Ma for the big garnet, which is further divided into 1828 (±11) Ma for the core and 1829 (±21) Ma for the rim domains. It is a pity that the MSWDs for these analyses were not shown, but I guess is that the core age has the lower one. Looking at these ages without errors, they are exactly what might be expected for the M1 and M2. Now the problem is the very large errors in the ages. To solve this the authors chose to use a pooled age of all the analyses to get smaller errors (result 1834 ± 17 Ma). This is a highly questionable way to use the data. It seems that the sampling was not structurally controlled to separate possible different garnet generations. To the S of Olkiluoto in the Turku area, two generation of garnets were described, the first elongated syn-D2 garnet was deformed in D3 which in turn was coeval with leucosome containing large garnets (Väisänen & Hölttä 1999; Fig. 14g). This might be the case in Olkiluoto, too. But the major problem here is not the sampling, but the resolution of the Lu-Hf garnet method, which is incapable of solving so detailed a problem.
You are spot on – Lu-Hf for these samples is not high enough resolution to split the older and younger events. You are also right that this is a quite new and not completely understood method (line 31 comment above). I have included the MSWDs as you suggested but the core MSWD is slightly higher than the rim. The reason for this is that MSWD evaluates the fit of the line to the data, the core data has a larger spread so it is more difficult to fit but results in a more well defined age (lower error). The rim data and the small garnet data have less spread in Lu/Hf ratios so the data is easy to fit a line to (low MSWD) but the age is more poorly defined (bigger error).

392: …all the analyses…
This has been accordingly modified.

394: …, then…
This has been accordingly modified.

396: these ages are not identical, just overlapping within error
This has been accordingly modified.

409: …the(se) data form…
This has been accordingly modified.

417: …in the U-Pb and…
This has been accordingly modified.

426: …The zircon U-Pb…
This has been accordingly modified.

475: Fig. 11. Why is the Turku granulite area skipped from the examples in Figure. It is anyway isotopically and metamorphically well-studied migmatite area close to Olkiluoto.
This reference, Väisänen & Hölttä 1999, has been added to the map thus also more constrains on P-T conditions in the Uusimaa belt.

499: …in the Häme Belt…
This has been accordingly modified.

509: "reaching slightly deeper crustal depth than the Häme Belt" ? what does this mean?
This statement has been re-written and modified, so that it infers deeper metamorphic depth.

512: …the Uusimaa Belt…
This has been accordingly modified.

513: …the anatectic melts (or melting)
This has been accordingly modified.

514-515: "indicates that the whole southern Finland domain was subjected to a long hot orogenic evolution with several crustal-scale melt pulses". Chardon et al. 2009. Tectonophysics 477, describes this nicely, worth citing
This reference has been added.

517: Chapter 5.3. This chapter shortly summarises the present knowledge of the chapter title, focusing on the shear zones, but it hard to understand how this literature review is connected to the data in the ms., garnet and apatite age determinations.
The coupling to the manuscript and reasoning to include this now re-written and information added so that this can be defined and tracked better.

519 and 521: back-arc vs backarc … Be consistent with writing style
This has been accordingly modified.

528: Additionally, additionally what? unclear
This statement has been re-written and modified.

531: …show that…
This has been accordingly modified.

574: "similarities between the Olkiluoto site and Häme Belt, and the differences to Uusimaa Belt." What actually are the real differences between the Häme and Uusimaa belts? So big that they are part of the main conclusions? This topic is not clearly discussed in the ms. There are evidently differences in erosion levels, but what else? If this refers to the metamorphic ages, two metamorphic events are found also in the Uusimaa Belt, 1.86 and 1.83 Ga (Väisänen et al. 2021, Inst.Seism., Univ Helsinki, Rep. S-71). The protolith ages are also the same, 1.86 Ga.

This part has been modified and main statement is that more studies is need in this region to better constrain the metamorphic history both for the Häme Belt and the Uusimaa Belt.

623: References: Many references occur twice in the list, e.g., Lahtinen et al. 2005, and many others. Delete the extra.
This has been fixed!

624: page numbers missing
This has been fixed!

714: page numbers missing. Check all the similar references
This has been fixed!

796: Pitkälä…This is an unpublished MSc thesis. A published document should be used: Pitkälä et al. 2018. Inst.Seism., Univ Helsinki, Rep. S-67.
This has been fixed!